# Molecular mechanism of toxin neutralization in the HipBST toxin-antitoxin system of *Legionella pneumophila*

Xiangkai Zhen[1,4], Yongyu Wu[1,4], Jinli Ge[2,4], Jiaqi Fu[3], Le Ye[1], Niannian Lin[1], Zhijie Huang[1], Zihe Liu[1], Zhao-qing Luo [3], Jiazhang Qiu[2] ✉ & Songying Ouyang [1] ✉

Toxin-antitoxin (TA) systems are ubiquitous genetic modules in bacteria and archaea. Here, we perform structural and biochemical characterization of the *Legionella pneumophila* effector Lpg2370, demonstrating that it is a Ser/Thr kinase. Together with two upstream genes, *lpg2370* constitutes the tripartite HipBST TA. Notably, the toxin Lpg2370 (HipT$_{Lp}$) and the antitoxin Lpg2369 (HipS$_{Lp}$) correspond to the C-terminus and N-terminus of HipA from HipBA TA, respectively. By determining crystal structures of autophosphorylated HipT$_{Lp}$, its complex with AMP-PNP, and the structure of HipT$_{Lp}$-HipS$_{Lp}$ complex, we identify residues in HipT$_{Lp}$ critical for ATP binding and those contributing to its interactions with HipS$_{Lp}$. Structural analysis reveals that HipS$_{Lp}$ binding induces a loop-to-helix shift in the P-loop of HipT$_{Lp}$, leading to the blockage of ATP binding and inhibition of the kinase activity. These findings establish the *L. pneumophila* effector Lpg2370 as the HipBST TA toxin and elucidate the molecular basis for HipT neutralization in HipBST TA.

Toxin–antitoxin (TA) systems are bacterial and archaeal genetic modules enriched in mobile genetic elements and chromosomes that comprise two or more closely linked genes encoding a toxin protein and its cognate antitoxin[1]. Since the discovery that the ccdB/ccdA TA system maintains stable inheritance of the mini-F plasmid in *Escherichia coli*[2], the biological roles of TA systems have been demonstrated to include maintaining stabilization and fitness of mobile genetic elements such as plasmids[3] and protection against phages[4]. Toxins are stable enzymes (e.g., RNases and kinases) or other proteins (e.g., gyrase inhibitors and pore-like toxins) that, in the absence of cognate antitoxin, interfere with vital cellular processes such as DNA replication and protein translation[5]. Antitoxins are unstable proteins or RNAs that counteract toxins. Based on the antitoxin nature and toxin-neutralization mechanism, TA systems can be divided into types I–VIII[6,7].

In type II TA systems such as HipBA modules, toxin neutralization depends on direct binding of a proteinaceous antitoxin[1,8]. HipA from HipBA TA module of the *E. coli* strain K-12 is a 440-amino acid (aa) Ser/Thr kinase that phosphorylates the tRNA^Glu-bound glutamate-tRNA ligase GltX at Ser239, thereby inhibiting protein translation[9,10]. The growth arrest induced by *E. coli* HipA can be counteracted by HipB, a cro/C1-type helix–turn–helix (HTH) domain-containing protein[11]. The structures of HipBA modules from *E. coli* (HipBA$_{Ec}$) and *Shewanella onesidensis* (HipBA$_{So}$) reveal that both TA modules form a HipA$_2$–HipB$_2$ heterotetramer in which HipB binds far from the kinase catalytic center of HipA[12]. Such neutralization strategy differs from most of other type II TA systems, where antitoxins binding usually occludes the active site[13] or mediate allosteric regulation of the toxin[14–16].

Recent studies demonstrated that TA systems containing toxins homologous to the *E. coli* HipA are widely distributed in bacterial

[1]Provincial University Key Laboratory of Cellular Stress Response and Metabolic Regulation, the Key Laboratory of Innate Immune Biology of Fujian Province, Biomedical Research Center of South China, Key Laboratory of OptoElectronic Science and Technology for Medicine of the Ministry of Education, College of Life Sciences, Fujian Normal University, 350117 Fuzhou, China. [2]State Key Laboratory for Zoonotic Diseases, College of Veterinary Medicine, Jilin University, Changchun, China. [3]Purdue Institute for Inflammation, Immunology and Infectious Disease and Department of Biological Sciences, Purdue University, West Lafayette, IN, USA. [4]These authors contributed equally: Xiangkai Zhen, Yongyu Wu, Jinli Ge. ✉e-mail: qiujz@jlu.edu.cn; ouyangsy@fjnu.edu.cn

genomes[17], suggesting the diversity of HipBA TA systems[17,18]. Among these TA systems, a tripartite system designated HipBST was recently identified and experimentally characterized in the enteropathogenic *E. coli* serotype O127:H6[18]. HipT, which serves as the toxin in the HipBST system, phosphorylates TrpS at Ser197, and its toxicity can be counteracted by the small protein encoded by the adjacent gene *hipS*[17]. Importantly, the toxin HipT and the antitoxin HipS of the HipBST system were found to correspond to the N-terminal subdomain 1 and the core kinase domain of the *E. coli* HipA, respectively. The third protein of the HipBST module, HipB, is analogous to HipB of the HipBA system and appears to enhance the neutralization effect of HipS by binding to an already formed HipT–HipS heterodimer[18]. Recently, a preprint study reported the structure of HipBST heterotrimer from *E. coli* serotype O127:H6 and concluded that ATP binding in HipT is prevented by comparing its structure in the heterotrimer to the available structures of *E. coli* HipBA complex[19]. However, the general mechanism for toxin neutralization in HipBST TA systems is not fully elucidated.

*Legionella pneumophila*, the causative agent of Legionnaires' disease, extensively modifies host signal transduction pathway, especially the post-translational modifications such as ubiquitination and phosphorylation, by translocating hundreds of effectors into the host cell via the Dot/Icm system[20–23]. One such effector is the recently identified Lpg2370[24], which was previously predicted to be an E3 ligase but has not been characterized[25].

In this work, we find that Lpg2370 in fact shares sequence identity with the C-terminus of the *E. coli* K-12 HipA, the toxin of type II TA system HipBA, which is then confirmed by experimental validation and determining the crystal structure of autophosphorylated Lpg2370 (pLpg2370). Furthermore, gene locus analysis indicates that *lpg2370* is grouped with *lpg2369* and *lpg2368* into a tricistronic operon, which we proceed to characterize respectively as the toxin HipT$_{Lp}$, the antitoxin HipS$_{Lp}$, and HipB$_{Lp}$, that constitute the tripartite TA system HipBST. We also determine high-resolution structures of the toxin pHipT$_{Lp}$ in complex with the ATP analog AMP–PNP and the binary complex with the antitoxin HipS$_{Lp}$ and identify key HipT$_{Lp}$ residues involved in ATP binding and interactions with HipS$_{Lp}$. Lastly, a comparison of the three structures determined in this study allowed us to propose the mechanism of toxin neutralization in the type II TA system HipBST.

## Results

### The *L. pneumophila* effector Lpg2370 is a Ser/Thr kinase
We first verified whether Lpg2370 is translocated into host cells via the type IV secretion system (T4SS) Dot/Icm as an effector protein. To this end, we performed TEM-1 β-lactamase translocation assay by infecting RAW264.7 macrophages with the fusion protein-expressing *L. pneumophila* cells grown to post-exponential phase. Vectors expressing TEM-1-RalF (positive control), TEM-1-FabI (negative control) or TEM-1-Lpg2370 fusion proteins were introduced into the T4SS-competent *L. pneumophila* strain Lp02 or the Dot/Icm-deficient strain Lp03, which were then assessed for the delivery of the β-lactamase fusions into the host macrophage cells by visual inspection under a fluorescence microscope. Cells infected by Lp02 cells expressing the TEM-RalF fusion protein emitted blue fluorescence, whereas infection with the TEM-FabI-expressing cells did not result in any emission of blue fluorescence by host cells (Fig. 1a). In addition, none of the fusion proteins were detectably translocated upon infection with the Dot/Icm-deficient strain Lp03 (Fig. 1a). Consistent with previous studies[24], the Lpg2370-overexpressing *L. pneumophila* strain Lp02 can be secreted into host cells, though at very low translocation efficiencies, suggesting that Lpg2370 is indeed a *L. pneumophila* effector protein.

Although previous studies implied that Lpg2370 is an E3 ubiquitin ligase based on sequence similarity with the RING-type E3 ubiquitin ligase FANCL[25–27], we repeated primary sequence analysis and did not find any significant similarity between the two proteins. However, we

found notable sequence identity (~20%) between Lpg2370 and the residues 64–440 of *E. coli* HipA, an atypical Ser/Thr kinase from the *E. coli* K-12 strain (Supplementary Fig. 1)[28]. The results particularly indicated conservation of the sequences corresponding to the P-loop (RISVAGAQ), the signature motif responsible for ATP binding, and the catalytic loop, which contains the catalytic residue D310 required for the kinase activity of *E. coli* HipA[29–31] (Fig. 1b). Moreover, comparing the Lpg2370 sequence to the NCBI database using basic local alignment search tool (BLAST) identified Lpg2370 as a HipA-like Ser/Thr kinase. These findings led us to hypothesize that Lpg2370 could be a kinase.

Kinases frequently undergo autophosphorylation on an invariable Ser or Thr residue in the P-loop[32]. For instance, autophosphorylation of Ser150 on the P-loop has been observed in HipA kinases from *E. coli* (HipA$_{Ec}$) and the proteobacterium *Shewanella oneidensis* (HipA$_{So}$)[30,31]. To investigate whether Lpg2370 is also autophosphorylated, we incubated purified recombinant Lpg2370 expressed in *E. coli* with the N6-benzyladenosine-5′-O-(3-thiotriphosphate) (N⁶-Bn-ATPγS) (Fig. 1c)[33]. Immunoblotting with anti-N⁶-Bn-ATPγS antibody detected a protein band corresponding to Lpg2370 (35 kDa) (Fig. 1d), which clearly indicated that Lpg2370 can be autophosphorylated. We next performed LC-MS/MS to identify the autophosphorylation site on Lpg2370. A mass shift of 79.97 Da ($m/z = 684.85$, $z = 2$) was consistently observed in the putative P-loop (53-MSVQGVQKK-61), revealing that the residue Ser54 within the P-loop is the autophosphorylation site (Fig. 1e). Taken together, these results suggest that Lpg2370 is a Ser/Thr kinase, though its substrates are currently unknown.

### Lpg2370 adopts a Ser/Thr kinase-like fold
To gain deeper insight into the molecular function of Lpg2370, we set out to determine its crystal structure. Diffraction phases for the SeMet-labeled Lpg2370 were determined using the single-wavelength anomalous diffraction method and the final structural mode was refined at 1.46 Å (Table 1).

Like other members of the protein kinase superfamily, Lpg2370 has a globular kinase fold that can be further divided into N-lobe and C-lobe. The N-lobe, which contains the P-loop, is composed of sheets β1–5 sandwiched by helices α1 and α2, whereas the C-lobe is predominantly α-helical and consists of helix bundles α3–α6 and α8–α11 and a short β-sheet β6–8 (Fig. 2a). In line with the results of primary sequence analysis, there is no apparent structural similarity between Lpg2370 and the E3 ligase FANCL (Supplementary Fig. 2a), and residues 4–28 of Lpg2370, which are relatively well aligned with FANCL[25], are a part of the typical kinase N-lobe. In agreement with our LC-MS/MS results, we can observe a phosphate group covalently attached to the Ser54 residue (Fig. 2b). However, unlike HipA$_{Ec}$ and HipA$_{So}$ whose P-loops are disordered upon serine autophosphorylation[29], the electron density of the Lpg2370 P-loop is well defined in the present structure (Fig. 2b). Interestingly, the P-loop of Lpg2370 differs from the counterparts of typical protein kinases in that it contains a single glycine residue[34]. The positioning of the phosphorylated Ser54 (pSer54) is stabilized by the positively charged side chains of K40, R131, and R134, as well as hydrogen bonds between side chain of Gln56 and side chains of Asp145 and Lys201 (Supplementary Fig. 2b).

Dali search suggested that Lpg2370 shares the highest structural homology score with HipA$_{Ec}$ and HipA$_{So}$ (Supplementary Table 1)[35]. Lpg2370 and HipA$_{Ec}$ superimpose with a relatively large root-mean-square deviation (RMSD) value of 3.846 Å over 221 Cα atoms (Fig. 2c, d). In addition to lacking a counterpart to the N-terminal region of HipA$_{Ec}$ (i.e., helices α1–α4 and strands β1–β3 of HipA), Lpg2370 differs from HipA$_{Ec}$ mainly in the configuration of the N-terminal lobe (Fig. 2c). One prominent difference is that the Lpg2370 P-loop with phosphorylated Ser54 is exposed to solvent in an orientation similar to that of unphosphorylated P-loop of HipA$_{Ec}$, which upon serine autophosphorylation rotates by ~180° and bends away from the

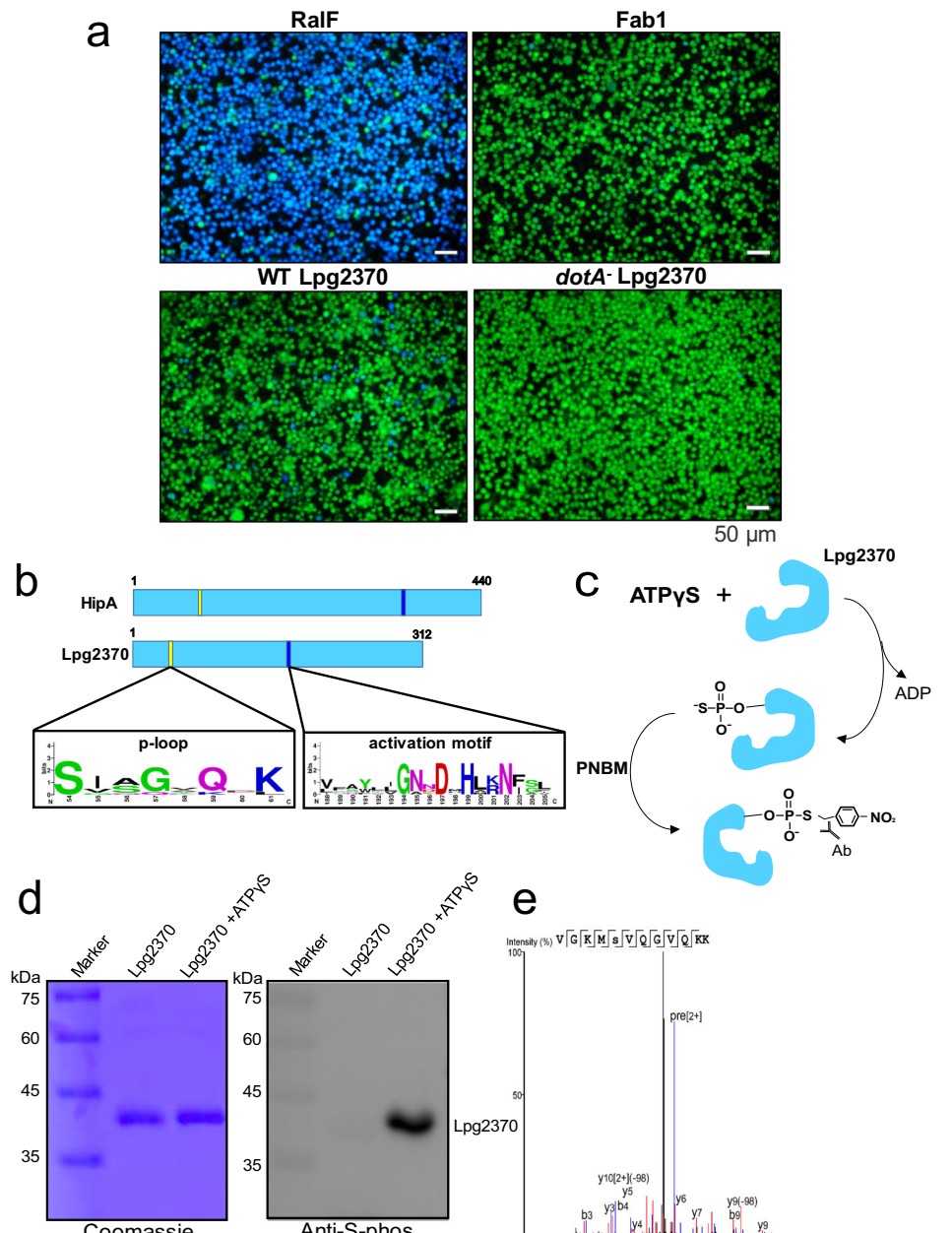

**Fig. 1 | *L. pneumophila* effector Lpg2370 is a Ser/Thr kinase. a** TEM-1 β-lactamase translocation assay demonstrates that Lpg2370 is a *L. pneumophila* effector protein. RAW264.7 cells were challenged with a T4SS-competent wild-type *L. pneumophila* strain Lp02 or the dotA-mutant deficient strain Lp03 carrying plasmids encoding TEM-1-RalF (positive control), TEM-1-FabI (negative control), or TEM-1-HipT$_{Lp}$. T4SS-mediated translocation of the fusion proteins into host cells was assessed 2 h after infection by the CCF4-AM-based fluorescence resonance energy transfer assay, scale bars 50 µm. **b** Schematic of Lpg2370 and the Ser/Thr kinase HipA from *E. coli* K-12 (HipA$_{Ec}$). Yellow and blue regions represent the approximate

locations of the P-loop and activation motif of HipA, respectively, as well as the locations of corresponding sequences in Lpg2370. Sequence conservation of P-loop and the activation motif is presented as a weblog below the protein schematics. **c** Diagram depicting detection of thiophosphorylated Lpg2370 by thiophosphate labeling. PNBM: p-nitrobenzylmesylate. **d** Thiophosphate labeling assay with the purified 6×His-tagged Lpg2370. Thiophosphorylated Lpg2370 was visualized by immunoblotting. **e** Identification of the phosphorylated peptide by LC-MS/MS. The peptide SVQGVQK was observed at charge state 2$^+$ in two forms differing by 70.97 Da in molecular mass.

N-terminal lobe by 17.3 Å (Fig. 2e, f). Conversely, the six C-terminal α-helices of Lpg2370 and HipA$_{Ec}$, including the catalytic residues and some ATP-binding residues, are almost perfectly aligned (Fig. 2e).

## Lpg2368−Lpg2369−Lpg2370 constitute the tripartite HipBST TA system of *L. pneumophila*

Further analysis of Lpg2370 showed that proteins containing the C-terminal domain of HipA are widespread in bacteria (Fig. 3a). Moreover, the structural similarity between Lpg2370 and *E. coli* HipA and the fact that *E. coli* HipA along with HipB from the same genomic

locus composes a type II TA system prompted us to examine the locus of *lpg2370*. Indeed, we found that *lpg2370* is preceded by open reading frames (ORFs) of *lpg2368* and *lpg2369*. Analogously to TA systems such as HipBA[28], *lpg2368*, and *lpg2369* as well as *lpg2369* and *lpg2370* overlap by 4 bp. Further analysis suggested that *lpg2369* encodes a 102-aa protein similar to the N-terminal region of *E. coli* HipA and that *lpg2368* encodes a 72-aa protein homologous to the helix−turn−helix (HTH) domain of HipB (Fig. 3b and Supplementary Fig. 3a, b). Such locus organization is reminiscent of the HipBST TA module in *E. coli* O127:H6[17,18], suggesting that the *lpg2368−lpg2369-lpg2370*

**Table 1 | X-ray data collection and refinement statistics**

| Dataset | pHipT$_{Lp}$ | pHipT$_{Lp}$–AMP–PNP | HipT$_{Lp}$–HipS$_{Lp}$ |
|---|---|---|---|
| **Data collection** | | | |
| Wavelength (Å) | 0.9792 | 0.9792 | 0.9792 |
| Space group | P2$_1$2$_1$2$_1$ | P2$_1$2$_1$2$_1$ | P4$_1$2$_1$2 |
| Cell dimensions | | | |
| a, b, c (Å) | 52.95, 64.64, 90.42 | 39.81, 86.31, 91.88 | 45.22, 45.22, 394.55 |
| α, β, γ (°) | 90.00, 90.00, 90.00 | 90.00, 90.00, 90.00 | 90.00, 90.00, 90.00 |
| Resolution range (Å) | 27.32–1.46 (1.51–1.46) | 36.52–1.59 (1.65–1.59) | 31.87–1.82 (1.885–1.82) |
| R$_{merge}$ | 0.125 (1.08) | 0.08 (1.22) | 0.084 (1.35) |
| CC1/2 | 0.95 (0.87) | 0.996 (0.870) | 0.999 (0.872) |
| I/σ (I) | 9.4 (2.00) | 10.3 (2.20) | 22 (2.60) |
| Completeness (%) | 95.32 (90.51) | 99.39 (98.12) | 99.9 (99.70) |
| Multiplicity | 8.57 (6.44) | 12.3 (9.80) | 24 (24.70) |
| **Refinement** | | | |
| Resolution (Å) | 27.32–1.46 (1.51–1.46) | 36.52–1.59 (1.65–1.59) | 31.87–1.82 (1.885–1.82) |
| R$_{work}$ (%) | 17.53 (24.15) | 17.20 (18.60) | 18.59 (27.23) |
| R$_{free}$ (%) | 19.42 (25.06) | 19.26 (21.40) | 19.28 (21.32) |
| Ramachandran plot (%) | | | |
| Favored region | 98.40 | 98.34 | 98.34 |
| Allowed region | 1.60 | 1.66 | 0.00 |
| Outliers region | 0.00 | 0.00 | 0.00 |

One crystal was used for the determination of each structure. Values in parentheses are for the highest-resolution shell.

locus is a potential tricistronic operon encoding component of a HipBST TA system.

In the HipBST TA system of *E. coli* O127:H6, the toxin HipT (denoted HipT$_{O127}$) can form a heterotrimeric complex with the anti-toxin HipS$_{O127}$ and the HTH domain protein HipB$_{O127}$[18]. We therefore performed size-exclusion chromatography, pull-down assays and isothermal titration calorimetry (ITC) to analyze interactions between Lpg2370, Lpg2369, and Lpg2368. The co-expressed 6×His-tag Lpg2369 and untagged Lpg2370 were co-eluted using Ni affinity chromatography, and the size-exclusion chromatography analysis revealed that the peak is shifted forward by 0.4 mL compared to the peak of Lpg2370 alone, suggesting Lpg2369 can interact with Lpg2370 (Fig. 3c). Moreover, size-exclusion chromatography indicated that Lpg2368 co-elutes with the co-expressed 6×His-tagged Lpg2369–Lpg2370 complex and binds to the Lpg2369–Lpg2370 complex assembled in vitro (Fig. 3c), which was then further confirmed by the pull-down assays (Supplementary Fig. 4). These results suggest that Lpg2370 directly interacts with Lpg2369, whereas Lpg2368 binds to a stable Lpg2369–Lpg2370 complex. Moreover, the results of ITC assays demonstrated that the dissociation constants between Lpg2370 and Lpg2369 and Lpg2370-Lpg2369 complex and Lpg2368 are 42 nM and 1.5 μM, respectively (Supplementary Fig. 5a, b), which is in agreement with the previously published data on HipBST$_{O127}$[18].

Given the established analogy between the *L. pneumophila* Lpg2368–Lpg2369–Lpg2370 operon and the HipBST TA system, we next aimed to functionally characterize Lpg2370 by investigating its potential toxicity to host bacteria. Heterogeneous expression of the recombinant Lpg2370 in *E. coli* had no observable effect on cell growth (Supplementary Fig. 6). The toxicity assays were performed in the *L. pneumophila* strain Lp02. To avoid undesirable effects from endogenous expression, we prepared deletion strain Δlpg2368-Δlpg2369-Δlpg2370 (Δ3) and examined bacterial growth upon overexpression of recombinant Lpg2370. Overexpression of Lpg2370 significantly

inhibited the growth of *L. pneumophila*, both on plates and in liquid medium (Fig. 3d). Moreover, the catalytically inactive H199A (H/A) mutant failed to inhibit bacterial growth (Fig. 3d), indicating that the kinase activity of Lpg2370 is strictly required for its toxicity (Fig. 3d).

To assess the impacts of Lpg2368 and Lpg2369 on bacterial growth, we inserted *lpg2370* into the low-copy-number IPTG-inducible vector pZL507, and *lpg2368*, *lpg2369*, or *lpg2368*–*lpg2369* were separately inserted into the plasmid pJL03 with the arabinose-inducible pBAD promoter. Growth and viability of the *L. pneumophila* Δ3 strain carrying combinations of these plasmids was then monitored. Growth inhibition caused by the expression of Lpg2370 was counteracted by co-expression of Lpg2369, suggesting that Lpg2369 functions as the antitoxin (Fig. 3e). Co-expression of Lpg2368 and Lpg2369 was also found to counteract Lpg2370-dependent growth inhibition, whereas the expression of Lpg2368 without Lpg2369 could not prevent the growth inhibition (Fig. 3e). Taken together, these results are consistent with the findings on the *E. coli* O127:H6 HipBST module[17,18] and demonstrate that Lpg2368, Lpg2369, and Lpg2370 from *L. pneumophila* constitute the tripartite HipBST TA system[18] and will thus hereafter be referred to as HipT$_{Lp}$, HipS$_{Lp}$, and HipB$_{Lp}$, respectively[17,18].

## The kinase activity of HipT$_{Lp}$ is likely independent of P-loop serine autophosphorylation

A comparison of the crystal structure of pHipT$_{Lp}$ and the structures deposited in the PDB revealed that the autophosphorylated P-loop in HipT$_{Lp}$ adopts an orientation similar to that of the P-loop in the crystal structure of *E. coli* HipA S150A mutant (Fig. 2f)[30]. This observation led us to speculate that pHipT$_{Lp}$ can bind ATP. Thermal shift assays performed with the purified wild-type HipT$_{Lp}$ revealed a 2.5 °C-increase in the melting temperature (Tm) in the presence of non-hydrolysable ATP analogue adenylyl-imidodiphosphate (AMP–PNP), suggesting that pHipT$_{Lp}$ indeed binds ATP (Supplementary Fig. 7). Likewise, isothermal calorimetry determined that the dissociation constant between pHipT$_{Lp}$ and AMP–PNP was about 70 μM (Fig. 4a), which is within the range of ATP-binding affinity expected for other kinases[29].

To elucidate how pHipT$_{Lp}$ binds ATP, we determined the crystal structure of pHipT$_{Lp}$ in complex with AMP–PNP at 1.36 Å resolution (Table 1). The structure of pHipT$_{Lp}$–ATP reveals that AMP–PNP is bound to the P-loop like in other representative kinases (Fig. 4b). The backbone of pHipT$_{Lp}$ in the complex is virtually identical to the apo structure (RMSD = 0.35 Å), with the exception of P-loop that bends towards helix α2 to accommodate the AMP–PNP. ATP (AMP–PNP)-interacting residues appear to be conserved among the bacterial HipT toxins, implying a shared mechanism for ATP binding (Fig. 4c). In pHipT$_{Lp}$, the γ-phosphate of AMP–PNP is stabilized by V58, H199, and D219, the β-phosphate forms hydrogen bonds with Q59, K61, and K85, whereas the α-phosphate interacts with K85 and N202. The adenosine moiety interacts with the main chain of K130 and forms π-stacking interactions with the side chain of F132 (Fig. 4b).

A previous study demonstrated that the kinase activity of *E. coli* HipA is essential for the growth arrest of host cells[36], and cell growth was inhibited when HipT$_{O127}$ was expressed in *E. coli* BL21 (DE3) cells[18]. To confirm the role of the residues involved into the ATP binding in the HipT toxins in vivo, HipT$_{O127}$ TA is used to perform the growth inhibition assays due to the easy manipulation of *E. coli* compared to *L. pneumophila*. To investigate whether the above-mentioned residues responsible for ATP binding are essential for the kinase activity of HipT, we performed in vivo toxicity assays with HipT$_{O127}$ variants in which residues corresponding to the S54 and the highly conserved ATP-binding residues of HipT$_{Lp}$ were substituted with alanine. Intriguingly, mutation on the residues corresponding to S54 of HipT$_{Lp}$ (S57A and S57D HipT$_{O127}$) remain toxic to *E. coli* cells, whereas substitutions of K64 (K61), K86 (K85), H212 (H199), N215 (N202), and D233 (D219) of HipT$_{O127}$ (corresponding residues in HipT$_{Lp}$ are indicated in parentheses) eliminated the toxic phenotype (Fig. 4d). Taken together,

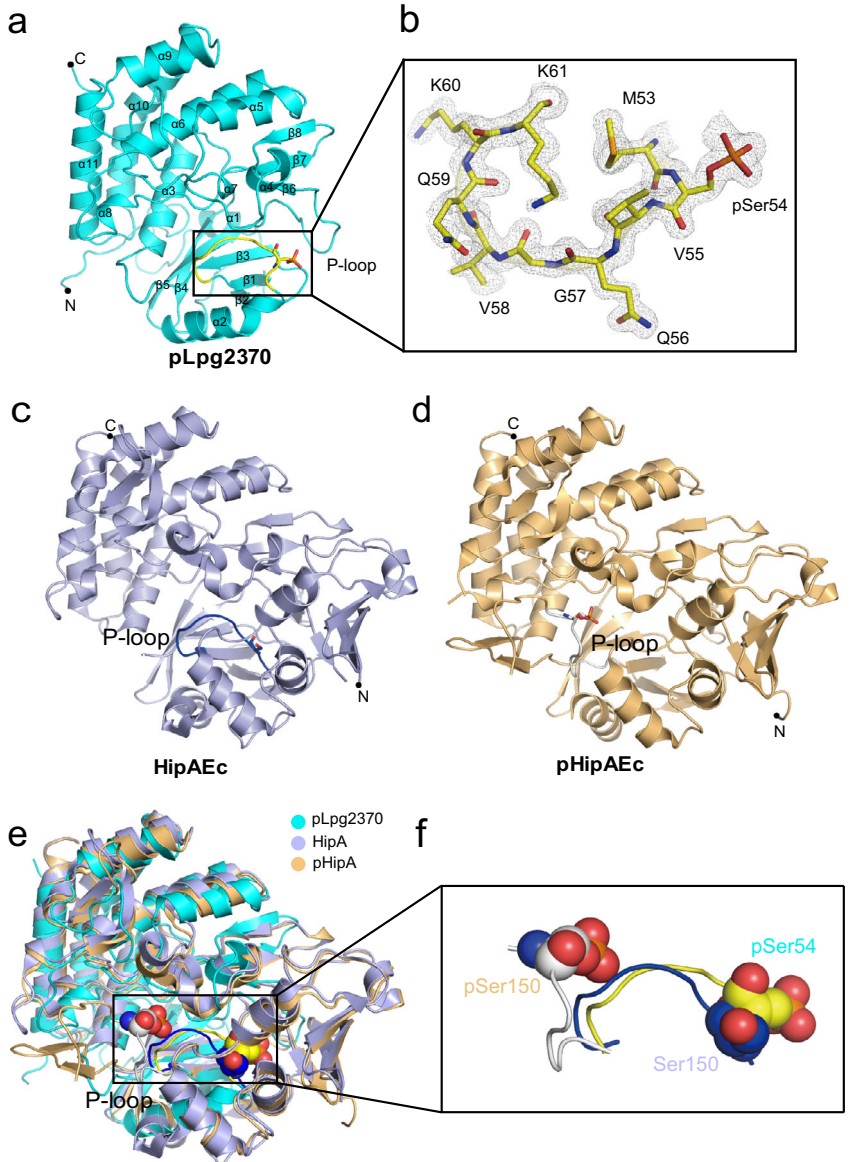

**Fig. 2 | Crystal structure of pLpg2370. a** Cartoon representation of pLpg2370. The N- and C-termini of pLpg2370 and its secondary elements of are labeled correspondingly. The P-loop is colored yellow and the pSer54 is shown in stick representation. **b** Detailed view of the P-loop with the pSer54. The 2Fo-Fc omit map is contoured at the 1.0 σ level and the P-loop residues are labeled. Crystal structures of the unphosphorylated HipA$_{Ec}$ (PDB ID: 3TPB) (**c**) and pHipA$_{Ec}$ (PDB ID: 3TPE) (**d**). The P-loops are labeled on both structures and colored blue and white, respectively. The P-loop serine/phosphoserine is shown as a stick representation. **e** Superimposition of pLpg2370 with the structures of pHipA and HipA. **f** Detailed view of the P-loops from the three superimposed structures. Ser150 in HipA$_{Ec}$ and phosphoserines in pHipA$_{So}$ and pLpg2370 are shown as spheres. Please note that the phosphorylated P-loop of Lpg2370 is bent toward N-lobe, similar to the unphosphorylated P-loop of HipA.

these results suggest that unlike in *E. coli* HipA, HipT retains the ATP-binding ability independent of the autophosphorylation on the conserved S54 in the P-loop and that HipT uses a universal mode for ATP recognition.

### Structural basis for the toxin HipT$_{Lp}$ recognition by the antitoxin HipS$_{Lp}$

Although the toxic activity of HipT in the HipBST TA system has been demonstrated to be counteracted by the antitoxin HipS[17,18], the underlying molecular mechanism remains unknown. We therefore sought to determine the structure of the HipT$_{Lp}$–HipS$_{Lp}$ complex. To express the HipT$_{Lp}$–HipS$_{Lp}$ complex, a ribosomal-binding site (RBS, AGGAGA)[37] was introduced between the stop codon of HipS$_{Lp}$ and the start codon of HipT$_{Lp}$. The resultant HipS$_{Lp}$–RBS-HipT$_{Lp}$ was cloned into pET21a (+) vector. The crystal structure of the SeMet-labeled

HipT$_{Lp}$–HipS$_{Lp}$ complex was determined and refined at 1.89 Å resolution (Table 1).

In the structure of HipT$_{Lp}$–HipS$_{Lp}$ complex, a copy of HipT$_{Lp}$ and HipS$_{Lp}$ each were observed per crystal asymmetric unit. Residues belonging to helices α1 and α2 of HipT$_{Lp}$ were not visible in the electron density map, whereas the density of the remaining residues was unambiguous (Fig. 5a). All 102 residues of HipS$_{Lp}$ were successfully built into the model, showing that HipS$_{Lp}$ is a small single-domain protein composed of five β-strands and three α-helices. The overall structure of the HipT$_{Lp}$-HipS$_{Lp}$ complex is highly similar to *E. coli* HipA, with HipS$_{Lp}$ and HipT$_{Lp}$ aligning with the N- and C-terminal portions of *E. coli* HipA, respectively (Supplementary Fig. 8a). HipS$_{Lp}$ superimposed with the N-terminus of *E. coli* HipA with an overall RMSD of 0.932 Å across 64 Cα. However, a notable difference can be observed on the β4–α2 loop of HipS$_{Lp}$, which is twisted and rotated by ~45° with

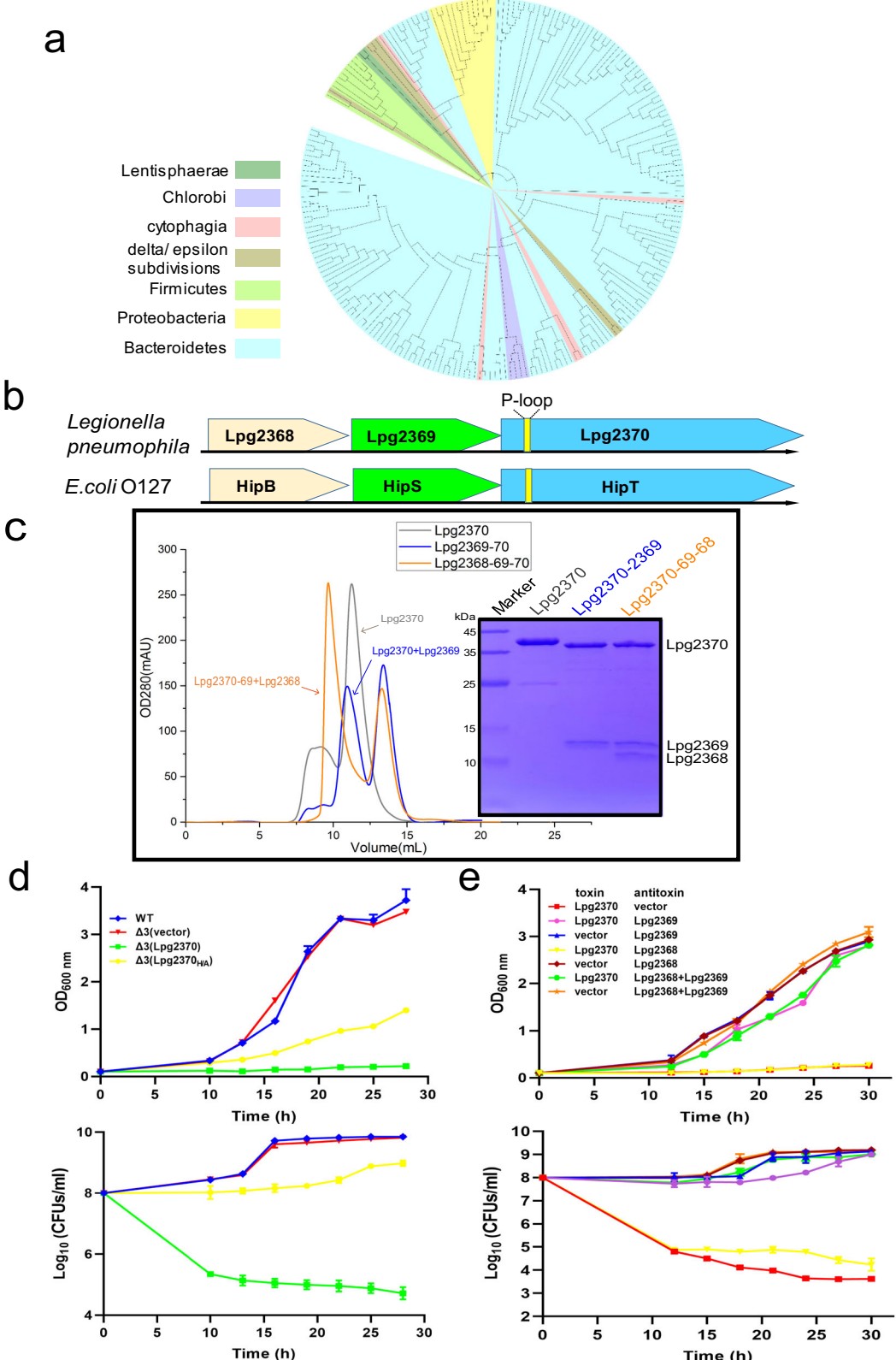

respect to its counterpart in the N-terminus of *E. coli* HipA (Supplementary Fig. 8b).

In the structure of HipT$_{Lp}$–HipS$_{Lp}$ complex, three α-helices of HipS$_{Lp}$ form a helix bundle that sits above the cleft formed by the β-sheet in the N-terminal lobe of HipT$_{Lp}$, whereas the β-strands form a flank region in HipS$_{Lp}$ (Fig. 5b). HipS$_{Lp}$ binds the toxin HipT$_{Lp}$ via hydrogen bonding in three main interacting regions, which constitute

more than 1100 $\text{Å}^2$ of total buried surface area (Fig. 5c–e). The intermolecular interactions are mainly formed between helices α1, α2 and α1–α2 loop of HipS$_{Lp}$ and helix α3 and strand β5 of HipT$_{Lp}$. In the first interacting region, side chains of HipS$_{Lp}$ E63 and HipT$_{Lp}$ K157 form a salt bridge, side chain of HipS$_{Lp}$ E58 engages in polar interactions with the main chain amide and side chain of HipT$_{Lp}$ R154, and hydrogen bonds are additionally formed between main chain of HipS$_{Lp}$ G59 and

**Fig. 3 | *L. pneumophila* genes *lpg2370*, *lpg2369*, and *lpg2368* belong to the same operon and constitute a tripartite HipBST family TA system. a** The HipA-C-terminal domain-like proteins are found in bacteria, branches of the IQTree maximum likelihood phylogenetic tree of representative HipA-C-terminal sequences are colored by the taxonomic groupings as per the upper left panel. **b** Schematic of the *lpg2368*–*lpg2369*-*lpg2370* operon in *L. pneumophila* and the HipBST operon from *E. coli* O127:H6 strain. In the genome of *L. pneumophila*, *lpg2368*, *lpg2369*, and *lpg2370* (relative position of the conserved P-loop is highlighted in yellow) are located in the same operon and encode proteins that share homology with the HipBST TA system. **c** Comparison of elution profiles of Lpg2370 (11.3 mL), Lpg2370-Lpg2369 (10.9 mL), and Lpg2370-Lpg2369-Lpg2368 (9.6 mL) in size-exclusion chromatography (Superdex 75 increase column). The accompanying SDS-PAGE gel with peak fraction samples is provided on the right side. **d** Growth curve (upper panel) and the CFU count (bottom panel) of *L. pneumophila* Δlpg2368-Δlpg2369-Δlpg2370 (Δ3) expressing the kinase site active mutant H199A (H/A) and the wild-type Lpg2370, Lpg2370 causes cellular growth arrest upon induction, whereas the kinase site active mutant H199A (H/A) does not. **e** Growth curve (upper panel) and the CFU count (bottom panel) of overnight cultures *L. pneumophila* Δlpg2368-Δlpg2369-Δlpg2370 (Δ3) harboring pZL507 (pZL507:: *lpg2370*) or the empty pZL507 combined with pJL03 (pJL03:: *lpg2368*), pJL03 (pJL03:: *lpg2369*), pJL03 (pJL03:: *lpg2368*–*lpg2369*) or the empty low-copy-number pJL03 vector, as indicated, the tested bacterial strains were diluted in fresh AYE broth (supplemented with 10 μg/mL Gentamicin) to $OD_{600} = 0.1$ and split into 2-mL subcultures. 200 μM IPTG was added to induce the expression of Lpg2370, and 1% arabinose was added to induce the expression of Lpg2368, Lpg2369, or Lpg2368 + Lpg2369. In panels **d** and **e**, data shown are mean values ± SD ($n = 3$ independent experiments).

side chain of $HipT_{Lp}$ K201 and main chains of $HipS_{Lp}$ I65 and $HipT_{Lp}$ G57 (Fig. 5c). The second interacting region includes a salt bridge between $HipT_{Lp}$ D77 and $HipS_{Lp}$ K73 and hydrogen bonds between (i) side chain of $HipT_{Lp}$ D133 and $HipT_{Lp}$ G94, (ii) side chain of $HipT_{Lp}$ R154 and side chain of $HipS_{Lp}$ N91 as well as main chain of $HipS_{Lp}$ V92, (iii) side chain of $HipS_{Lp}$ N91 and main chain of $HipT_{Lp}$ Y79, and (iv) side chains of $HipT_{Lp}$ Q78 and $HipS_{Lp}$ Q90 (Fig. 5d). In the third interacting region, the side chain of $HipT_{Lp}$ Q148 hydrogen bonds with the main chain of $HipS_{Lp}$ F56, whereas the side chain of $HipT_{Lp}$ E144 forms hydrogen bonds with the side chain of $HipS_{Lp}$ S38 and the main chain of $HipS_{Lp}$ L39 (Fig. 5e).

To verify the importance of these interactions for stable binding of $HipS_{Lp}$ to $HipT_{Lp}$, we performed pull-down assays with untagged wild-type $HipS_{Lp}$ and wild-type or mutant $HipT_{Lp}$ carrying a N-terminal 6×His-tag. The $HipT_{Lp}$ mutants D133A, R134A, and E144A completely lost their ability to bind $HipS_{Lp}$ and the mutants R154A, K157A, K201A exhibited severely reduced $HipS_{Lp}$ binding, suggesting that these residues form key interactions with $HipS_{Lp}$ (Fig. 5f).

## Molecular mechanism for toxin neutralization in the HipBST TA systems

One of the most striking features of the HipBST TA systems is that the role of antitoxin is taken by HipS which corresponds to the N-terminal portion of HipA toxin from the *E. coli* HipBA system. To better understand how the toxic activity of HipT is neutralized by HipS, we reinspected and compared the structures of apo $pHipT_{Lp}$, $pHipT_{Lp}$–AMP–PNP complex, and $HipT_{Lp}$–$HipS_{Lp}$ complex. Apo $pHipT_{Lp}$ and $HipT_{Lp}$ from the $HipT_{Lp}$–$HipS_{Lp}$ complex superimpose with RMSD of 0.464 Å over 215 Cα atoms. Notably, Ser54 of $HipT_{Lp}$ is phosphorylated in the structure of apo $pHipT_{Lp}$ but not in the $HipT_{Lp}$–$HipS_{Lp}$ complex (please note that $HipT_{Lp}$–$HipS_{Lp}$ was co-expressed in *E. coli* BL21) (Fig. 2a, b and Supplementary Fig. 8c). Since the residue S54 is phosphorylated when $HipT_{Lp}$ is expressed alone, we wondered whether the phosphorylation on S54 influences the interaction between $HipT_{Lp}$-$HipS_{Lp}$ and $HipB_{Lp}$. Size-exclusion chromatography revealed that the phosphorylation state of S54 does not appear to have a noticeable effect on interactions between $HipT_{Lp}$-$HipS_{Lp}$ and $HipB_{Lp}$ (Supplementary Fig. 9). Moreover, structural comparison suggests that the P-loop of $HipT_{Lp}$, which encircles ATP and is critical for catalytic activities in typical Ser/Thr kinases, underwent a conformational change from loop to helix upon $HipS_{Lp}$ binding (Fig. 6a, b). Such allosteric regulation induced by the antitoxin binding has not been observed in *E. coli* HipBA TA system[29]. A conformational change similar to the loop-to-helix change of the $HipT_{Lp}$ P-loop in $HipT_{Lp}$-$HipS_{Lp}$ can also be observed in the recently released structure of $HipBST_{O127}$ trimer[19] (Supplementary Fig. 10), suggesting a common mechanism of toxin neutralization.

These observations led us to hypothesize that the loop-to-helix conformational transition induced upon $HipS_{Lp}$ binding may obstruct the access of ATP to the kinase active site, resulting in inhibition of the $HipT_{Lp}$ kinase activity. Superimposition of the structures of $pHipT_{Lp}$,

$pHipT_{Lp}$–AMP–PNP, and the $HipT_{Lp}$–$HipS_{Lp}$ further revealed that P-loop in the $HipT_{Lp}$–$HipS_{Lp}$ complex overlaps with AMP–PNP in the $pHipT_{Lp}$–AMP–PNP complex (Fig. 6c, d and Supplementary Fig. 11). More specifically, the γ-phosphate and β-phosphate groups of AMP–PNP would clash with the side chains of Q59 and D219, respectively, whereas the α-phosphate would clash with the side chains of K61 and K85 (Fig. 6d). This may account for the unphosphorylated state of the P-loop in the $HipT_{Lp}$–$HipS_{Lp}$ complex when they were co-expression (Supplementary Fig. 8c). Such conformational change also occurs in the P-loop of $HipBST_{O127}$, suggesting that a similar mechanism is utilized by $HipBST_{O127}$[19] (Fig. 6e, f). To further verify whether ATP binding is abolished, we measured the binding affinity between the $HipT_{Lp}$–$HipS_{Lp}$ complex and AMP–PNP with ITC and found that $HipT_{Lp}$ completely lost the AMP–PNP binding affinity when binding with $HipS_{Lp}$ (Fig. 6g). Consistent with these results, the thermal stability of the $HipT_{Lp}$-$HipS_{Lp}$ complex did not change upon the addition of 4 mM AMP–PNP (Supplementary Fig. 11a). Considering that $HipA_{Ec}$ in autophosphorylated form can bind ADP and AMP but not ATP[30], we also investigated whether the $HipT_{Lp}$–$HipS_{Lp}$ complex binds ADP and AMP. Again, the results of ITC experiments suggested that the $HipT_{Lp}$–$HipS_{Lp}$ complex has no detectable affinity for ADP or AMP, even at concentrations of 1 mM (Supplementary Fig. 12). The allosteric regulation of the P-loop induced by the antitoxin binding was also observed in the $HipBST_{O127}$[19]. Together, these findings suggest that $HipS_{Lp}$ binding induces conformational changes in the P-loop of $HipT_{Lp}$, which blocks ATP binding and consequently inhibits the $HipT_{Lp}$ kinase activity (Fig. 6h).

## Discussion

Although the biological functions of TA systems are often ambiguous and debatable, recent advances in the field and the discovery of numerous novel TA modules increasingly support their roles in viral defense or plasmid stability[4,7] and interactions between hosts and their mobile genetic elements. Among known TA modules, the type II TA systems are most well understood. Type II TA systems typically consist of two components, though several tripartite type II TA modules have been identified, such as the *Rv1955-Rv1956-Rv1957* TA-chaperone (TAC) system of *Mycobacterium tuberculosis*[38,39], and the recently discovered *E. coli* HipA-like TA system HipBST of *E. coli* O127: H6[18].

Most type II antitoxins are composed of an N-terminal DNA-binding domain that regulates transcription from the TA locus through direct interaction with the promoter, and a C-terminal region responsible for the toxin binding and inhibition[40]. HipBA modules are representative type II TA systems and are ubiquitous in bacteria[28]. Based on the similarity searching using *E. coli* HipA against the sequenced microbial genomes, a three-component widespread HipBST TA was found and experimentally verified[18]. Later bioinformatic analysis has suggested the presence of several other HipBA-like TA in numerous bacterial species[17]. While the toxin $HipT_{O127}$ of HipBST[O127] was found to exert its toxic function by phosphorylating TrpS, and in spite of the similarity between $HipB_{O127}$ and *E. coli* HipB,

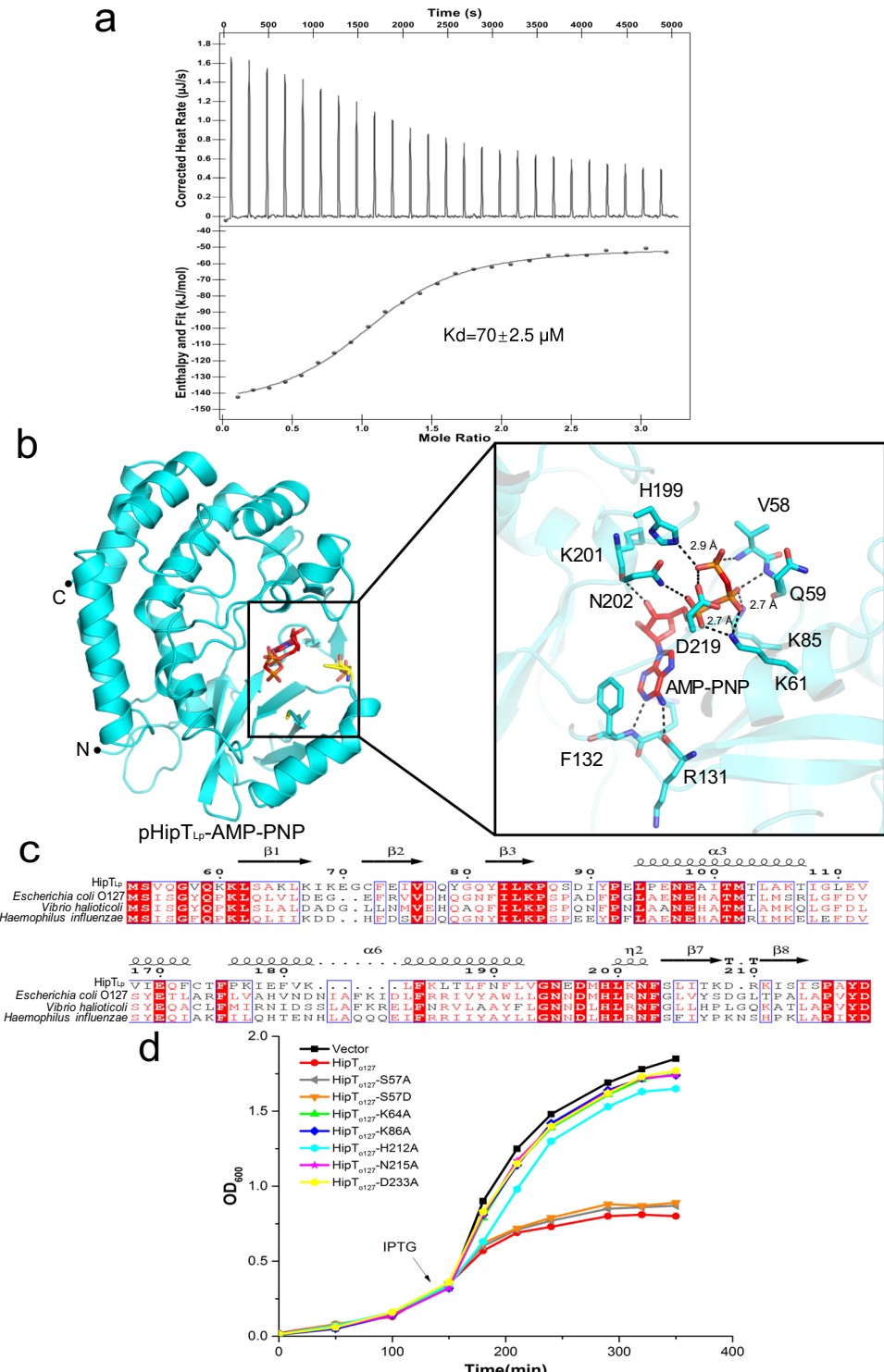

**Fig. 4 | Crystal structure of the pHipT$_{Lp}$–AMP–PNP complex. a** Binding of AMP–PNP to pHipT$_{Lp}$ monitored by ITC, the presented data is from a single ITC experiment. **b** Left: cartoon representation of the pHipT$_{Lp}$–AMP–PNP complex. The N- and C-termini of pHipT$_{Lp}$ are labeled. The bound AMP–PNP molecule and pSer54 are shown as sticks and colored red and yellow, respectively. Right: detailed view of the ATP-binding cavity of HipT$_{Lp}$ and interactions formed with AMP–PNP. The distance between the interacting residues of HipT$_{Lp}$ and AMP–PNP are in range of 2.7–3.3 Å, which was shown as sticks and hydrogen bonds are indicated with black dashed lines. **c** Sequence alignment of HipT variants from *L. pneumophila*, *Escherichia coli* O127:H6, *Vibrio halioticoli* and *Haemophilus influenzae* reveals conservation of the residues involved in ATP binding. The ATP-binding residues are encircled with black bold rectangles. **d** Growth curves of *E. coli* BL21(DE3) cells expressing recombinant wild-type HipT$_{O127}$ or its mutant variants S57A, S57D, K64A, K86A, H212A, N215A, and D233A.

HipB$_{O127}$ however cannot neutralize the HipT$_{O127}$ kinase. This task was taken over by HipS, a small protein with homology to the N-terminal part of *E. coli* HipA[18,19]. Our current study also experimentally identifies a tripartite HipBST$_{Lp}$ TA in which the antitoxin HipS$_{Lp}$ instead of HipB$_{Lp}$ restores the growth inhibition induced by HipTLp in human pathogen *L. pneumophila*, which is in agreement with the HipBST$_{O127}$ TA[18].

Together with the preprint study in ref. [19], our study identifies and elucidates the toxin-neutralization mechanism in the tripartite

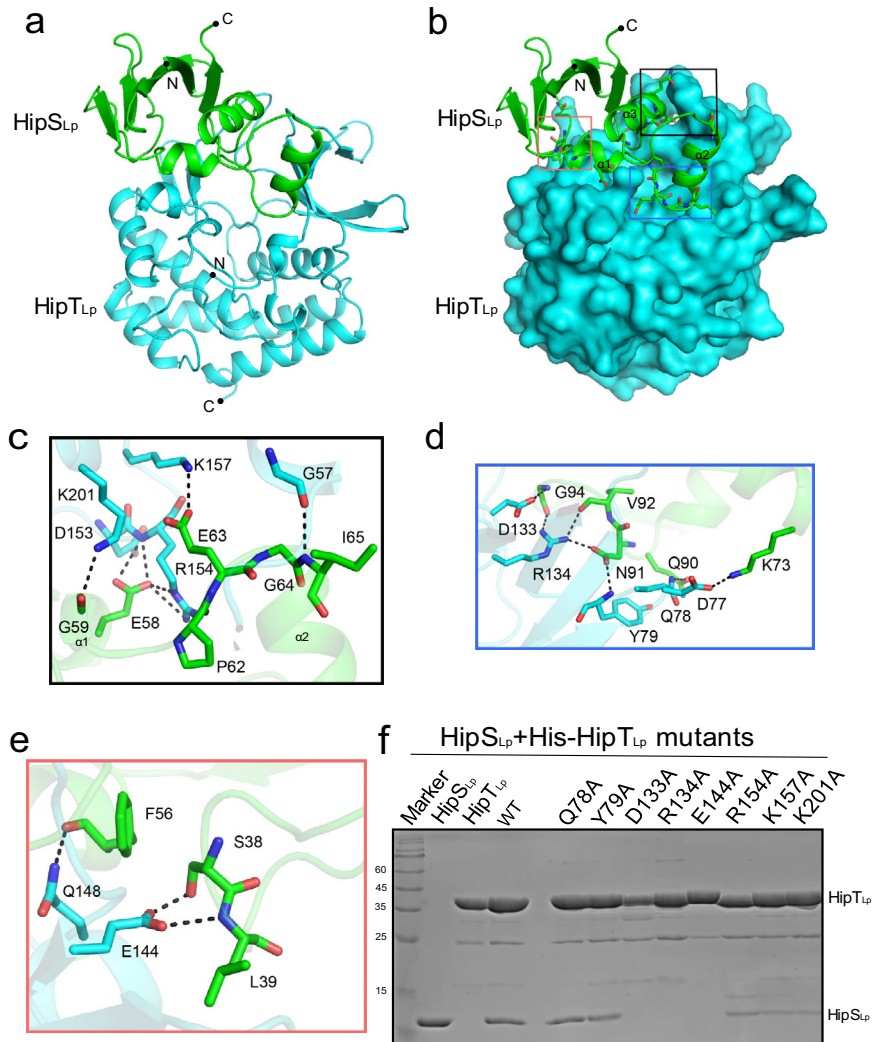

**Fig. 5 | Structural basis for HipT$_{Lp}$ recognition by HipS$_{Lp}$. a** Cartoon representation of the HipT$_{Lp}$–HipS$_{Lp}$ complex. HipT$_{Lp}$ and HipS$_{Lp}$ are colored cyan and green, respectively, and their N- and C-termini are labeled correspondingly. **b** Relative locations of three interacting regions formed between HipT$_{Lp}$ and HipS$_{Lp}$. The two proteins are color-coded as in panel A, and HipT$_{Lp}$ is shown in surface representation. Interacting residues of HipS$_{Lp}$ are shown as sticks and the three regions involved in interaction with HipS$_{Lp}$ are encircled with black, blue, and red rectangles, respectively. **c**–**e** Detailed view of the three interacting regions formed between HipT$_{Lp}$ and HipS$_{Lp}$. Interacting residues are shown as sticks and hydrogen bonds are indicated with black dashed lines. **f** Pull-down assays performed with wild-type or mutant HipT$_{Lp}$ carrying C-terminal 6×His-tag and untagged wild-type HipS$_{Lp}$.

HipBST TA systems of *L. pneumophila*. It is clear that the toxin neutralization mechanism in HipBST systems is notably different from the corresponding mechanism in HipBA TA system of *E. coli*[18]. The structural study by Bærentsen et al. on the *E. coli* O127:H6 HipBST found that HipT adopts an inactive conformation in the HipBST complex that prevents ATP binding[19]. However, it was unclear whether the blockage of ATP binding arises from the binding of the antitoxin HipS or autophosphorylation of the conserved P-loop serine[19]. Our structure of the HipT$_{Lp}$–HipS$_{Lp}$ complex clearly shows the toxicity of HipT$_{Lp}$ is neutralized directly upon HipS$_{Lp}$ binding, leading to blocking of the ATP-binding site through steric hindrance. Although this study demonstrates that HipB$_{Lp}$ forms a heterotrimer with the HipT$_{Lp}$–HipS$_{Lp}$ complex, we failed to obtain crystals of the heterotrimer structure after extensive crystal screening. However, the available structure of *E. coli* HipBST shows that HipB binds to HipT but does not interact with the kinase active site[19]. In addition, overall architecture of HipBST$_{O127}$ system is reminiscent of the structure of HipBA$_{So}$ system in the HipBA$_{So}$-DNA complex, implying that HipBST can also bind DNA[19]. Induction of HipB$_{O127}$ was found to be required and sufficient for the transcriptional repression of the HipBST TA[19]. Thus, it

seems that the toxin neutralization and the autoregulation of the HipBST TA are carried out by the antitoxin HipS and HipB, respectively[19]. Nevertheless, the exact role of HipB in the HipBST systems remains to be determined.

To the best of our knowledge, HipT$_{Lp}$ is the only identified toxin of TA systems secreted into host cells via the type IV secretion system Dot/Icm. Nevertheless, the employment of TA toxins by the pathogenic bacteria during host cell infection is not unprecedented. For instance, the type III secretion system effector AvrRxo1-ORF1 from *Xanthomonas oryzae* pv. *oryzicola* constitutes a type II TA system with adjacent AvrRxo1-ORF2[41]. AvrRxo1 can phosphorylate NAD in planta, leading to the suppression of the flg22-triggered ROS burst[42]. Moreover, previous investigation of fourteen effectors from *Burkholderia gladioli* revealed that the restriction endonuclease Tox-Rease-5 domain-containing effector TaseT$_{Bg}$ can be employed to inhibit the growth of co-habiting bacterial species, and its activity is counteracted by the associated immunity protein TsiTBg[43]. Apart from the effectors LegK1–K5 and LegK7, HipT$_{Lp}$ is the sixth effector known to function as a Ser/Thr kinase in *L. pneumophila*. LegK1 phosphorylates NF-κB to activate the genes expression involved in inflammation during

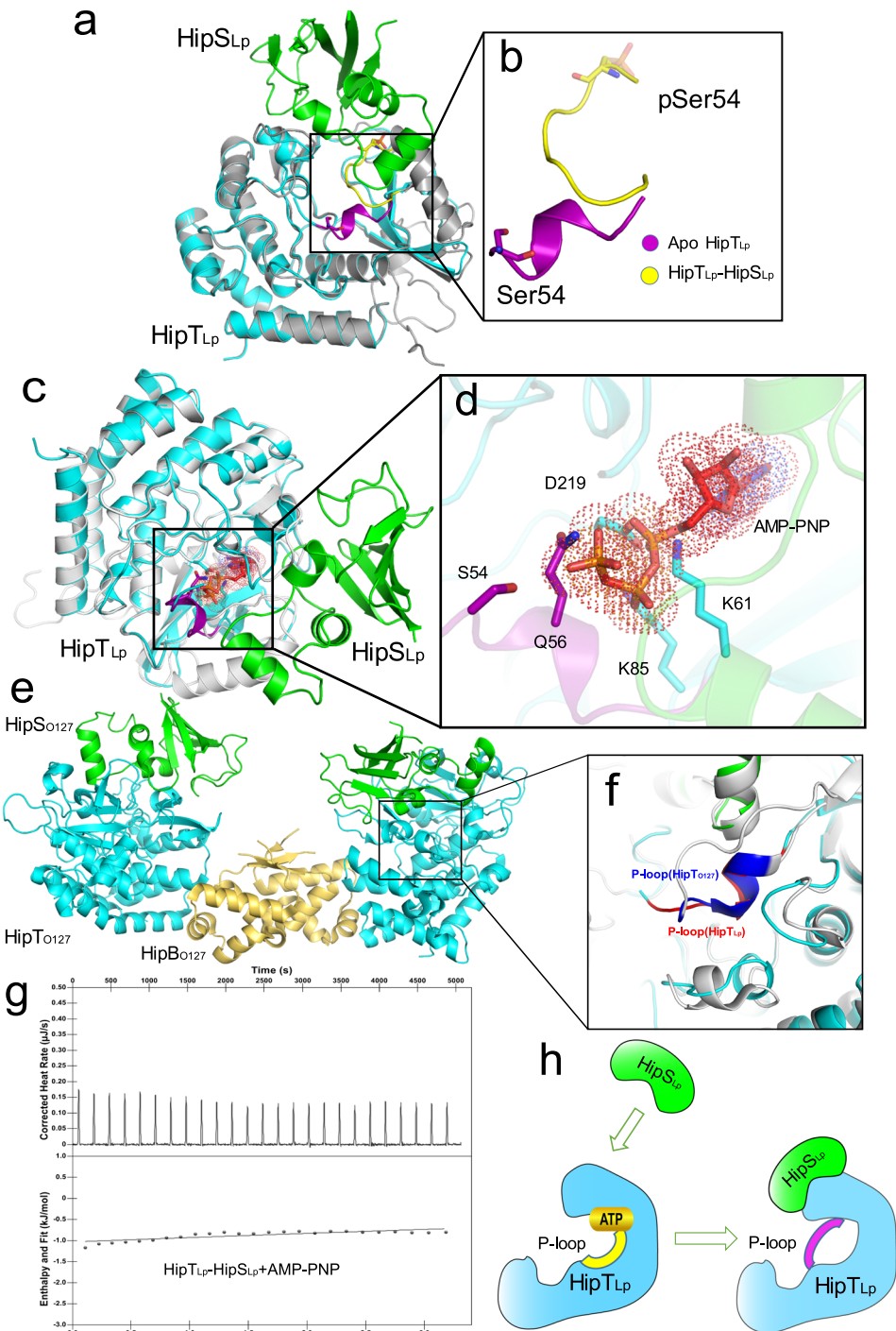

**Fig. 6 | Toxin-neutralization mechanism in the HipBST TA systems. a** Overlay of apo pHipT$_{Lp}$ and structure of HipT$_{Lp}$–HipS$_{Lp}$ in cartoon representation. Apo pHipT$_{Lp}$ is colored gray and its P-loop is colored yellow, whereas HipT$_{Lp}$ from the binary complex is colored cyan and its P-loop is colored purple. HipS$_{Lp}$ is colored green. **b** Close-up view of the overlay showing that HipS$_{Lp}$ binding induces conformational change of the P-loop. **c** Overlay of the pHipT$_{Lp}$–AMP–PNP and the structure of HipT$_{Lp}$–HipS$_{Lp}$ in cartoon representation. pHipT$_{Lp}$ from the pHipT$_{Lp}$–AMP–PNP is colored white. The HipT$_{Lp}$–HipS$_{Lp}$ complex is color-coded as in panel **a**. **d** Close-up view of the overlapping between HipT$_{Lp}$ residues and AMP–PNP. AMP–PNP is shown as dotted surface mode and overlapping HipT$_{Lp}$ residues are shown as sticks. **e** Crystal structure of HipBST$_{O127}$ (PDB:7AB5) HipT$_{O127}$, HipS$_{O127}$ and HipB$_{O127}$ were colored cyan, green and yellow orange, respectively. **f** Structural comparison between the HipT$_{Lp}$–HipS$_{Lp}$ and HipBST$_{O127}$. The P-loops of HipT$_{Lp}$ and HipT$_{O127}$ are colored red and blue, respectively. **g** ITC thermogram and binding curve demonstrated that the HipT$_{Lp}$–HipS$_{Lp}$ complex does not display detectable affinity for AMP–PNP. **h** Proposed model for toxin-neutralization mechanism in the HipBST TA systems. The toxin HipT is a Ser/Thr kinase in which the P-loop motif is vital for ATP binding and subsequent substrate phosphorylation. Binding of the antitoxin HipS causes conformational changes in the P-loop, which blocks ATP binding and ultimately inhibits the kinase activity of HipT.

infection[44]. Phosphorylation of Hsp70 at T495 by LegK4[45], leading to the inhibition of protein synthesis. LegK7 phosphorylates a conserved scaffold protein MoB1, hijacking the hippo pathway to promote its survival[33,46]. The exact substrate of HipT$_{Lp}$ in its host remains to be identified in future studies. In summary, these results suggest that TA systems could serve as a reservoir for additional secreted effectors, which sheds light on the evolutionary links between the TA system and the effectors secreted by the pathogenic microorganisms.

## Methods

### Bacterial strains and growth conditions

The *L. pneumophila* strain Philadelphia-1 derivative Lp02[47] was used as the progenitor of all derivative strains used in this study. To construct *Δlpg2368*-*Δlpg2369*-*Δlpg2370* triple-deletion mutant strain of *L. pneumophila* (termed Δ3), we first constructed deletion plasmid by cloning the upstream and downstream flanking regions into pSR47S. Briefly, the 1.2-kb fragment located upstream of *lpg2368* and the 1.2-kb fragment located downstream of *lpg2370* were obtained by PCR using Δ3 A1/Δ3 A2 and Δ3 B1/Δ3 B2 primer pairs and high-fidelity FastPfu DNA polymerase (TransGen) (Supplementary Table 2), respectively. The amplified PCR products were used as templates to produce a DNA fragment containing the flanking regions by fusion PCR with primers Δ3 A1/Δ3 B2 and high-fidelity FastPfu DNA polymerase (TransGen). After digestion with BamHI and SalI, the DNA fragment was inserted into BamHI/SalI-digested pSR47S[48]. The deletion plasmid was then introduced into Lp02 by triparental mating, and conjugants were selected on CYET plates containing kanamycin (20 μg/mL) and streptomycin (50 μg/mL). Deletion mutants were verified by standard colony PCR techniques using primers Δ3 A1/Δ3 B2 and 2×Taq master mix (Novoprotein) from colonies grown on CYET plates containing 5% sucrose[49]. Moreover, the *lpg2370*, *lpg2368*, *lpg2369*, and *lpg2368–lpg2369* genes were inserted into pZL507 or pJL03 as BamHI/SalI fragments to construct pZL507-Lpg2370, pJL03-Lpg2368, pJL03-Lpg2369, and pJL03-Lpg2368-Lpg2369 plasmids, which were then introduced into the *Δlpg2368Δlpg2369Δlpg2370* strain. For effector translocation, fresh single colonies of Lp02 or Lp03 harboring expression plasmid for TEM-1-Lpg2370, TEM-1-RalF, or TEM-1-FabI fusion proteins were streaked onto BCYE plates 2 days before infection.

### TEM-1 β-lactamase translocation assays

To test Dot/Icm-dependent transfer of the fusion proteins into host cells, *L. pneumophila* cells expressing the fusion proteins were grown in the presence of 0.5 mM IPTG to post-exponential phase and used to infect monolayers of RAW264.7 cells that were seeded in 96-well plates at an MOI of 20. The CCF4-AM substrates (Invitrogen, Carlsbad, CA) were mixed with medium in the wells two hours after infection. After further incubation for 1 hour at room temperature, infected cells were inspected under a Nikon IX-80 fluorescence microscope equipped with a β-lactamase FL-Cube (U-N41031, Chroma Technology Corp, Bellows Falls, VT). Images of infected cells were obtained using a DP-72 color fluorescence camera (Olympus). Translocation of the β-lactamase chimeras was assessed by the presence of cells emitting blue fluorescence signals. The percentage of infected cells was determined by counting the number of cells emitting blue fluorescence in specified areas of the wells. TEM-1-RalF and TEM-1-FabI fusion proteins were used as positive and negative controls, respectively. Experiments were performed in triplicate, and at least 300 cells were counted in each sample.

### Protein expression

DNA fragments encoding full-length HipT$_{Lp}$, HipS$_{Lp}$, and HipB$_{Lp}$ or their variants were inserted into pET21a (+) containing a N-terminal 6×His-tag, respectively. To obtain HipT$_{Lp}$–HipS$_{Lp}$ complex, HipS$_{Lp}$–HipT$_{Lp}$ was cloned in the pET21a (+) following previously described methods[37,50] to express the 6×His HipT$_{Lp}$–HipS$_{Lp}$. The recombinant plasmids were transformed into *E. coli* BL21 (DE3) cells for protein expression. Overexpression was induced in log phase cultures (OD$_{600}$ ≈ 0.6) by cooling cultures on ice for 20 min and adding IPTG to a final concentration of 0.4 mM, followed by overnight incubation at 16 °C and 220 rpm. The overexpression of the wild-type HipT$_{Lp}$ did not lead to the growth arrest in *E. coli*. After target protein expression, cells were pelleted by centrifugation and resuspended in buffer A (50 mM Tris-HCl, pH 8.0, 100 mM NaCl). The cells were then lysed by ultrasonication and the lysate was centrifuged at 17,000×*g*

and 4 °C for 30 min. The supernatant was loaded onto Ni$^{2+}$-NTA column (Qiagen) for purification of target recombinant proteins. After washing with 100 mL of buffer A supplemented with 50 mM imidazole, the target proteins were eluted with buffer A supplemented with 250 mM imidazole. Fractions containing the target protein were pooled, concentrated to 0.5 mL and then purified with Superdex 75 increase column (GE Healthcare) equilibrated with buffer B (20 mM Tris-HCl, pH 8.0, 150 mM NaCl).

Selenomethionine (SeMet)-labeled HipT$_{Lp}$ and HipT$_{Lp}$–HipS$_{Lp}$ were expressed in M9 medium supplemented with 2 mM MgSO$_4$, 0.1 mM CaCl$_2$, 0.5% w/v glucose, 2 mg/L biotin, 2 mg/L thiamine, and 0.03 mg/L FeSO$_4$. When bacterial cultures reached OD$_{600}$ of 1.0, final concentrations of 100 mg/mL of phenylalanine, lysine, and threonine, and 50 mg/mL of isoleucine, leucine, valine, and of SeMet (Chemie Brunschwig) were added in form of solid powder, after which the cultures were incubated for 30 min. Protein expression was then induced with 0.2 mM IPTG, and the cultures were further incubated on a shaker at 16 °C for 20 h. Cells were collected at 5000×*g* for 15 min and 4 °C and resuspended in the lysis buffer (50 mM Tris pH 8.0, 100 mM NaCl, 5 mM β-mercaptoethanol). Protein purification was performed as described above.

### In vitro kinase assays

Purified Lpg2370 (1 μg) was resolved by SDS-PAGE and transferred to the PVDF membrane using a Bio-Rad wet transfer system. The membrane was blocked with 5% milk for 1 h at room temperature. The membrane was incubated overnight at 4 °C with a primary antibody against thiophosphate ester (rabbit anti-thiophosphate ester (ab92570), Abcam, dilution 1:4000 v/v). The membrane was then washed three times with TBST before being incubated with a secondary antibody (HRP-conjugated AffiniPure goat anti-rabbit (Cat No. SA00001-2), Proteintech, dilution 1:5000 v/v) for 1 h at room temperature. The membrane was washed three times with TBST, and proteins were detected using an ECL detection reagent.

### Liquid chromatography-mass spectrometry (LC-MS) analysis

LC-MS was used to analyze autophosphorylation of purified recombinant Lpg2370. After staining of gels with Coomassie blue, excised gel segments were subjected to in-gel trypsin digestion and dried. Electrospray ionization mass spectrometry (ESI-MS) was performed using an integrated HPLC/ESI-MS system (1260 Infinity, Agilent Technologies/amaZon SL, Bruker Corporation) equipped with a Luna 5 μm C18 column (100 Å, 250 × 4.60 mm, 5 μm). Peptides were dissolved in 10 μl 0.1% formic acid and were auto-sampled directly onto a homemade C18 column (35 cm × 75 μm i.d., 1.9 μm 100 Å). Samples were then eluted for 60 mins with linear gradients of 3–35% acetonitrile in 0.1% formic acid at a flow rate of 300 nl/min. The mass spectrometer was equipped with a CaptiveSpray source. Survey scans were recorded over 100–1700 *m/z* range and the mass spectra data were acquired with a timsTOF Pro mass spectrometer (Brucker) operated in PASEF mode. PASEF setting: 10 MS/MS scans (total cycle time 1.27 s), charge range 0–5, active exclusion for 0.4 min, Scheduling Target intensity 10000, Intensity threshold 2500, CID collision energy 42 eV.

The raw files generated from LC-MS/MS were analyzed by Peaks Studio X software (Bioinformatics Solutions Inc., Waterloo, ON, Canada) against the input of the Lpg2370 amino acid sequence. The following database search criteria were set to: enzyme, trypsin; variable modification, phosphorylation, precursor ion mass tolerance, 10 ppm; MS/MS fragment mass tolerance, 0.02 Da; tryptic enzyme specificity with two missed cleavages allowed. Identifications were filtered according to mass accuracy and 1% false discovery rate.

### Protein crystallization and collection of crystallographic data

The purified target protein was concentrated to 0.5 mL and loaded onto a Superdex 75 increase column (GE Healthcare) Fractions

containing purified proteins were then concentrated at 4000×*g*, 4 °C to ~15 mg/mL using an Amicon Ultra 30 K centrifugal filter. To obtain HipT$_{Lp}$–AMP–PNP complex, the purified HipT$_{Lp}$ was incubated with AMP–PNP at a 1:1.2 molar ratio at 4 °C for 30 min and concentrated to about 12 mg/mL.

For crystallization of SeMet-labeled HipT$_{Lp}$, HipT$_{Lp}$–AMP–PNP complex, and SeMet-labeled HipT$_{Lp}$–HipS$_{Lp}$ complex, the purified and concentrated protein samples were mixed with the reservoir solution at equal volumes and crystallized using the sitting drop vapor diffusion method at 16 °C. Initial crystals of SeMet-labeled HipT$_{Lp}$ and HipT$_{Lp}$–AMP–PNP complex were obtained within three days in condition containing 8% Tacsimate (pH 6.0) and 20% w/v PEG 3350. Initial crystals of the SeMet-labeled HipT$_{Lp}$-HipS$_{Lp}$ complex were obtained in the condition containing 0.1 M sodium acetate (pH 7.0) and 12% v/w PEG 3350. After extensive optimization, diffraction-quality crystals of SeMet-labeled HipT$_{Lp}$ and the HipT$_{Lp}$–AMP–PNP complex were grown in the presence of 10% Tacsimate (pH 6.2) and 20% v/w PEG 3350. Diffraction-quality crystals of the HipT$_{Lp}$–HipS$_{Lp}$ complex were grown in the presence of 0.1 M sodium acetate (pH 7.2) and 15% w/v PEG 3350. Harvested crystals were preserved in the respective reservoir solutions supplemented with cryoprotectant and flash-frozen in liquid nitrogen.

### Structure determination and refinement
All X-ray diffraction data were collected at the BL-02U1 station of the Shanghai Synchrotron Radiation Facility (SSRF). Single-wavelength anomalous diffraction (SAD) datasets of SeMet-labeled HipT$_{Lp}$ and SeMet-labeled HipT$_{Lp}$–HipS$_{Lp}$ complex were obtained at high resolution and the data were processed with the HKL-2000 package[51]. Autosol program of PHENIX package was used for SAD phasing and initial model building. Residues 72–312 of HipT$_{Lp}$ were auto-built, whereas the residues 1–71 residues were built by iterative manual building in Coot[52]. Structure refinement was carried out with PHENIX[53]. The HipT$_{Lp}$–AMP–PNP binary complex was determined using the molecular replacement method with the structure of HipT$_{Lp}$ as the search model. Structure quality was analyzed during PHENIX refinements and later validated in the PDB validation server. Detailed crystallographic and structure refinement data are listed in Table 1. Structural images were generated using PyMol (Schrödinger, LLC).

### Mutagenesis
Base substitutions in this study were introduced using two pairs of complementary primers (sense and antisense strand primers) containing the desired mutation. The primers used in this study were listed in Supplementary Table 2. All constructs were verified by DNA sequencing.

### Isothermal titration calorimetry (ITC)
ITC experiments were performed in Nano ITC Low Volume (TA instruments). All samples were prepared in the buffer containing 20 mM HEPES (pH 8.0) and 150 mM NaCl. Typically, the titrant concentration in the syringe was 200–500 μM, and the titrand concentration in the reaction cell was 10–20 μM. Titration was conducted at 25 °C using multiple injection method with 150 s intervals. Obtained data were integrated, corrected, and analyzed using the NanoAnalyze software (TA Instruments) with a single-site binding model.

### Pull-down assays
To perform pull-down assays, wild-type or mutant HipT$_{Lp}$ carrying C-terminal 6×His-tag were incubated with Ni-agarose beads for 30 min and then washed twice with buffer containing 20 mM Tris-HCl (pH 8.0) and 150 mM NaCl. The beads were then incubated with untagged wild-type HipS$_{Lp}$ (6×His-tag was previously cleaved with TEV protease) for 1 h and then washed twice. The proteins were eluted from beads using a buffer containing 20 mM Tris-HCl (pH 8.0), 150 mM NaCl, and 250 mM imidazole. Eluted samples were analyzed using SDS-PAGE analysis.

### In vivo toxicity assays
For in vivo toxicity assays of HipT from *E. coli* serotype O127:H6 (denoted HipT$_{O127}$) to *E. coli*, which was performed by expressing the *hipT$_{O127}$* gene or its mutant variants in *E. coli* BL21(DE3). The gene encoding HipT$_{O127}$ was synthesized and cloned into pET21a (+) vector. Plasmids encoding HipT$_{O127}$ single-point mutants S57A, S57D, K64A, K86A, H212A, N215A, and D233A were prepared using site-specific mutagenesis, and the primers are listed in Supplementary Table 2. Transformed *E. coli* BL21 (DE3) cells were plated on agar and a single bacterial colony was transferred to 10 mL LB medium for culturing. Expression of wild-type HipT$_{O127}$ and its mutant variants was induced with IPTG at 0.2 mM concertation when the bacterial cultures reached OD$_{600}$ = 0.5, after which the bacterial growth curve was measured every 30 min for 6 h.

For the Lpg2370 in vivo toxicity assays in *L. pneumophila*, Lpg2370, Lpg2370-Lpg2369, Lpg2370-Lpg2368, Lpg2370-Lpg2369-Lpg2368 were overexpressed in *L. pneumophila* Lp02 or the Δ*lpg2368*Δ*lpg2369*Δ*lpg2370* deletion strain. Overnight cultures of the tested bacterial strains were diluted in fresh AYE broth to OD$_{600}$ = 0.1 and split into 2-mL subcultures into which different concentrations of arabinose or IPTG were added. The subcultures were then grown at 37 °C with constant rotation at 180 rpm. Cell viability was assessed by readout of OD$_{600}$ value every 3 h and plotting the values on a log scale.

### Thermal shift assays
Thermal shift assays were performed using 1 mg/mL phosphorylated HipT$_{Lp}$ or its mutants incubated with ATP/AMP–PNP and varied concentrations of nucleotides (0–4 mM) in 50 mM Tris-HCl (pH 8.0) and 150 mM NaCl. Then the mixture was loaded in 96-well PCR plates, the fluorescence signals were recorded as a function of temperature using Prometheus NT.48 (NanoTemper Technologies) in FRET mode. Fluorescence intensity was measured at Ex/Em of 350/330 nm. The temperature gradient range was set as 20–95 °C with a 0.5 °C ramp over the course of 30 s. Control assays were conducted in the same buffer without ATP/AMP–PNP. The thermal unfolding value (Tm) for pHipT$_{Lp}$ was calculated using the curve fitting software PR.Therm-Control (NanoTemper Technologies).

### Phylogenetic analysis
The sequence of Lpg2370 was blasted in the Uniprot (https://www.uniprot.org/) and all the hit sequences were used for sequence alignment in ClustalW, which was used to build the phylogenetic tree using MEGA software[54]. The phylogenetic tree was visualized using iTOL[55].

### Statistics and reproducibility
The TEM-1 β-lactamase translocation assays experiments in Fig. 1a, the western blotting experiment in Fig. 1d, the pull-down assays in Figs. 3c, 5f, Supplementary Fig. 4, and Supplementary Fig. 9 were performed at least triplicate at two independent times.

## Data availability
The atomic coordinates and structure factors of the autophosphorylated toxin HipT$_{Lp}$, the complex of HipT$_{Lp}$ with AMP–PNP, and HipT$_{Lp}$-HipS$_{Lp}$ binary complex have been deposited in the Protein Data Bank under the accession codes 7VKC, 7WCF, and 7VKB. Source data are provided with this paper.

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

## Acknowledgements

This work was supported by the National Key Research and Development Program of China (2021YFC2301403 to S.O), the National Nature Science Foundation of China grants (32170045 to X.K.Z., 82172287 to S.O., 31970134 and 32170182 to J.Z.Q.), the Fujian Provincial Department of Science and Technology (2020Y4007 to S.O.), and the High-level personnel introduction grant of Fujian Normal University (Z0210509 to S.O.). The diffraction data were collected at the beamline BL-02U1 of the Shanghai Synchrotron Radiation Facility (SSRF). We thank the support of the scientific research innovation program "Xiyuanjiang River Scholarship" of the College of Life Sciences, Fujian Normal University (22FSSK003 to X.K.Z.). We also thank our colleague Vanja Perčulija for the scientific and language editing of the manuscript.

## Author contributions

X.K.Z., S.O., and J.Z.Q. conceived the project and designed the experiments, X.K.Z. and Y.Y.W. performed crystallization and resolved the structures, J.L.G. performed the in vivo toxicity assays, J.Q.F., L.Y., N.N.L., Z.J.H., and Z.H.L. contributed to the protein expression and purification. X.K.Z. analyzed the structures, X.K.Z., S.O., Z.Q.L., and J.Z.Q. designed the biochemical experiments and wrote the manuscript.

## Competing interests

The authors declare no competing interests.
