## [Peer Review File · Nature Communications]

Editorial Note: This manuscript has been previously reviewed at another journal that is not operating a transparent peer review scheme. This document only contains reviewer comments and rebuttal letters for versions considered at Nature Communications .

[Mentions of prior referee reports have been redacted.]

REVIEWER COMMENTS

Reviewer #2 (Remarks to the Author):

The authors have addressed most of my comments well. However, there remain a few issues:

1. Ref 4, Guegler, C. K. & Laub, M. T. Shutoff of host transcription triggers a toxin-antitoxin system to cleave phage RNA and abort infection. Mol Cell, doi:10.1016/j.molcel.2021.03.027 (2021), was not removed as requested and as their Response indicates. This ref provides no new mechanism for phage inhibition; it should be removed. The same mechanism was published 26 years earlier in the 1996 ref., which established the field of phage inhibition by TAs.

2. 1 38: There are no reputable data relating TAs and antibiotic tolerance/persistence. Delete passage and ref. 6. Changing “persistence” to “antibiotic tolerance” is not an improvement.

3. I don't see where this information was added to the revised text to prevent confusion:

“1 102 and throughout: were the deletion mutants verified by sequencing?”

Response: We apologize for omitting this important piece of information. The deletion methods of *L. pneumophila* is as used in the ref 53. *L. pneumophila* strains containing in frame deletion were constructed through allelic exchange using a conditionally replicating plasmid pSR47s having the Kan R and sacB (sucrose sensitivity) markers. Sequences adjacent to the start and stop codons of each gene were amplified in two PCR reactions. These were then cloned on the suicide plasmid pSR47s (kanRsacB) to introduced into *L. pneumophila* by triparental mating. After selecting for integration by plating on kanamycin-containing medium, transconjugants were purified and plated onto 5% sucrose CYET plates to select for sucrose resistant recombinants that had lost the plasmid, using PCR analysis with appropriate primer pairs to determine if the deletions had been isolated.”

4. No, the fact that HipT is non-toxic in *E. coli* must occur earlier in the text as originally requested. No change was made by the authors.

Reviewer #3 (Remarks to the Author):

This work extends the field's knowledge on a "tripartite" Hip-like system found in bacterial chromosomes. Members of this family have previously been described with some characterization, including available crystal structures; however, nothing to date is available for this system from *L. pneumophila*. While much of the current findings are complementary to existing reports, the authors use crystal structures and biochemical assays to demonstrate the autophosphorylation of HipTLg only in the absence of the HipSLg binding partner. Further, they demonstrate that HipSLg is sufficient to block toxicity of HipTLg. In a somewhat strange experiment they switch to HipTEcO127 to demonstrate that auto-phosphorylation is not required for toxicity. Lastly, this manuscript reports the significant and unique finding that the HipTLg toxin is an effector for the Dot/Icm T4SS. Overall, this work reports advances of interest to the field of both TA systems and T4SS for an important human pathogen.

Items that must be addressed

1. One critical item must be addressed, and that is to clarify how the deletion mutants were verified. Whole genome sequencing is by far the preferred method; however, Sanger sequencing in combination with complementation (as was carried out) could be sufficient.
2. In addition, the size exclusion data for analyzing which protein(s) interact with others is unclear, as no sizes or calibrations are indicated. Presenting each sample alone, then adding in other samples, would clarify this. As it stands, no sound conclusions can be drawn from the data as shown.

Strongly suggested changes

3. Include the organism name in the title to direct readers between different systems but with the same names.
4. One of the most exciting findings is the use of this toxin as an effector; please move Fig S11 into the main text, and consider adding this information to the manuscript title.
5. A pervasive statement throughout the manuscript (starting in the abstract), is that the HipT and HipS sequences “derive from” HipA. However, there is no data to support which direction evolution has gone, if at all – it is incorrect to imply one arose from the other or vice versa. It is worth remarking that they share consensus domain structure, sequence, and apparent function, but it should not be phrased with any specific relationship.
6. The manuscript is more difficult to follow due to nomenclature of the proteins of interest. It would be helpful to establish early in the manuscript that Lg2370 is HipT, Lg2369 is HipS, and Lg2368 is HipB, then refer to them by these more descriptive names (with the Lg subscript). (I had to use a “cheat sheet” with 70=toxin, etc, to help me understand the manuscript).

Other comments

Abstract

7. Line 17-18: based on the modifications of the text, I recommend changing “have been proposed” to “have been demonstrated”, as the data behind those findings are rarely (ever?) questioned. Also, perhaps you can drop “important”, leaving it as TA systems have “roles”.

8. Line 21-25: recommend splitting this into two (or more) sentences expressing one main idea each

Text

9. Line 35: it might be worth noting the TA systems are enriched in mobile genetic elements.

10. Line 45: remove “in normal growth conditions” (not sure what that would be!)

11. Line 49: the idea that Type II systems are most prevalent is likely heavily biased by what has been searched for, so perhaps this should be phrased as “most studied”, or “most focused on” or something similar

12. Line 50: needs a citation. Is HipA really the most prevalent?

13. Line 61: please add “of the toxin” after the statement about allosteric regulation to clarify you are not referring to any action of transcription repression (conditional cooperativity)

14. Line 66: do both references 22 and 23 apply to this specific work discussed?

15. Line 70: do not say “derived from” unless there is evidence this is the case

16. Line 74: some of these sections are phrased strangely. The mechanism of toxin neutralization may have been unknown until now, but it is phrased as if it remains unknown today. The strangeness of this is because it is what is being reported, and therefore should not be considered unknown anymore.

17. Please ensure the BioRxiv paper is discussed in this section and the presented results are framed in relationship to what is reported. I appreciate that it is now included in the Discussion, but should really also appear in the Introduction section

18. Lines 81-82: another strange phrase of “characterization is still pending” while this manuscript is reporting this characterization, so it is not “still pending”.

19. Line 82: remove the word “detectable” (for example, the opposite is not true – there is no such phrase as “undetectable sequence identity”)

20. Line 83: and throughout please check the use of “terminal” and “terminus”. I think in this line you are referring to it as a noun, so the “terminus”; if referring to a location it would be “terminal”.

21. Line 113: If HipA is the most prevalent family, why are the comparisons limited to the E coli HipA?

22. Line 131: These results demonstrate this is a Ser/Thr kinase, but are limited to auto-phosphorylation based on the presented data. I appreciate that the phosphorylated substrates may remain unknown, but I am not sure you can state it is a functioning Ser/Thr kinase without showing this functional data. Please clarify this in the text.

23. Line 145: Why then would a prediction rely on such a small match, and why does this segment match FANCL?

24. Line 148-149: are the resolutions of these compared structures comparable? It is easy to image a higher resolution structure such as in the current work would show more ordered regions.

25. Line 168: this is a bit clunky “HipA C-terminal domain-containing proteins”. Can you come up with a better way to express this?

26. Line 187: Based on these data you cannot exclude Lg2368 binding to either individual protein unless you specifically tested this. (See note below about the Figure for these data)
27. Line 203: Why show the delta2 data with H/A mutant then?
28. Line 206: could you clarify? Is this the three individual open reading frames, and then the 68-69 ORFs together?
29. Line 208: just a typo, “0” instead of “delta” symbol
30. Line 212: What does “revert” convey?
31. Line 213: Why is this, “in contrast”? I do not think I understood the meaning of “revert” in the previous sentence. Does this indicate that 68+69 are toxic when expressed? If so, how does this fit the model of toxin-antitoxin?
32. Line 220: please clarify these are conformations in crystals, leaving open a possibility (even if unlikely) that in solution it may be different
33. It seems very strange / out of place / that mid-way through the experimental system is switched to E coli 0127 for toxicity / mutagenesis. Can the authors provide some transitional statement as to why Ec was more appropriate?
34. Line 261: does the BioRxiv paper have a structure of the complex?
35. Line 264: Why is it necessary to insert an RBS for the toxin gene? Is it to ensure stoichiometrically equal expression?
36. Line 305: see note above about assigning origins of HipA versus HipT-HipS

37. Line 310-315: If the toxin alone is phosphorylated, and the toxin co-expressed with the antitoxin is not phosphorylated, why would this lead to questions about HipB? Please clarify which samples were used for SEC that allows this conclusion to be drawn.

38. Line 319: this seems unlikely to be a coincidence. Comparing a T-S complex to a B-A complex seems incorrect, since T-S is homologous to HipA.

39. Line 422: just a small typo, should be “identified” (not “unidentified”)

Fig 3

40. Panel b, what is the yellow line denoting on the toxin genes?

41. Panel c, this is confusing. Which fractions are on the gel? Which eluted peaks correspond to what? Without information about molecular weight and column calibration (or comparison to the elution of individual proteins), we cannot follow any conclusions drawn from these data. I cannot understand the interaction of HipB from these data.

42. Panel d, e, why does this His to Ala mutation block toxicity? (e.g., what is the catalytic or functional role of this His?). These panels are way too small; it is difficult to distinguish the individual lines in panel e. Why is the delta3 (Lpg2370+vector) shown three times? Are these replicates? Why not show some average?

Fig 4

43. Panel a, how many replicates?

44. Panel b, should clarify what bond lengths or ranges were used to assign these dashed lines as bonds.

Fig 6

45. In general the wording in the legend seems off; I also suggest you drop “complex” from the description, as you’ve already established that HipT and HipS are bound together.

46. Panel g, I believe the top graph is called a “thermogram” rather than a “binding curve”?

Reviewer #2 (Remarks to the Author):

The authors have addressed most of my comments well. However, there remain a few issues:

1. Ref 4, Guegler, C. K. & Laub, M. T. Shutoff of host transcription triggers a toxin-antitoxin system to cleave phage RNA and abort infection. *Mol Cell*, doi:10.1016/j.molcel.2021.03.027 (2021), was not removed as requested and as their Response indicates. This ref provides no new mechanism for phage inhibition; it should be removed. The same mechanism was published 26 years earlier in the 1996 ref., which established the field of phage inhibition by TAs.

Response: The authors apologize for not removing reference 4. We made sure that reference 4 is removed in the revised manuscript.

2. I 38: There are no reputable data relating TAs and antibiotic tolerance/persistence. Delete passage and ref. 6. Changing “persistence” to “antibiotic tolerance” is not an improvement.

Response: Reference 6 and the passage on the role of TA systems in antibiotic tolerance was deleted.

3. I don't see where this information was added to the revised text to prevent confusion:

Response: We apologize for omitting this important piece of information. The information has been added to lines 397–407 of the revised manuscript: “To construct Δ lpg2368- Δ lpg2369- Δ lpg2370 triple deletion mutant strain of *L. pneumophila* (termed Δ 3), we first constructed deletion plasmid by cloning the upstream and downstream flanking regions into pSR47S. Briefly, the 1.2 kb fragment located upstream of *lpg2368* and the 1.2 kb fragment located downstream of *lpg2370* were obtained by PCR using Δ 3 A1/ Δ 3 A2 and Δ 3 B1/ Δ 3 B2 primer pairs (Table S2), respectively. The amplified PCR products were used as templates to produce a DNA fragment containing the flanking regions by fusion PCR with primers Δ 3 A1/ Δ 3 B2. After digestion with BamHI and Sall, the DNA fragment was inserted into BamHI/Sall-digested pSR47S⁵⁰. The deletion plasmid was then introduced into Lp02 by triparental mating, and conjugants were selected on CYET plates containing kanamycin (20 μ g/mL) and streptomycin (50 μ g/mL). Deletion mutants were verified by standard colony PCR using primers Δ 3 A1/ Δ 3 B2 from colonies grown on CYET plates containing 5% sucrose⁵¹”.

“I 102 and throughout: were the deletion mutants verified by sequencing?”

Response: Deletion mutants were verified by standard colony PCR method using primer pairs complementary to flanking regions of the deleted genes. This is a commonly used method for verifying knockout of bacterial genes. The method is illustrated in the figure below.

Colony PCR using primers A1 and B2

If deletion was NOT successful, the PCR product would be 2.4 kb plus the size of deleted genes

If deletion was successful, the PCR product would be 2.4 kb

4. No, the fact that HipT is non-toxic in *E. coli* must occur earlier in the text as originally requested. No change was made by the authors.

Response: The authors apologize for unintended negligence. In fact, initially, we did not realize that Lpg2370 is a HipA-like toxin, and we noticed that the over-expression Lpg2370 was not toxic to *E. coli* when the wild-type Lpg2370 was purified in *E. coli* to perform kinase activity assays *in vitro* and the LC-MS/MS. So, to explain why we use the *L. pneumophila* to perform the toxicity assay, a sentence stating that Lpg2370 is not toxic in *E. coli* is now added to lines 192–193 of the revised manuscript (Fig.s6).

Reviewer #3 (Remarks to the Author):

This work extends the field's knowledge on a "tripartite" Hip-like system found in bacterial chromosomes. Members of this family have previously been described with some characterization, including available crystal structures; however, nothing to date is available for this system from *L. pneumophila*. While much of the current findings are complementary to existing reports, the authors use crystal structures and biochemical assays to demonstrate the autophosphorylation of HipTLg only in the absence of the HipSLg binding partner. Further, they demonstrate that HipSLg is sufficient to block toxicity of HipTLg. In a somewhat strange experiment they switch to HipTEcO127 to demonstrate that auto-phosphorylation is not required for toxicity. Lastly, this manuscript reports the significant and unique finding that the HipTLg toxin is an effector for the Dot/Icm T4SS. Overall, this work reports advances of interest to the field of both TA systems and T4SS for an important human pathogen.

Items that must be addressed

1. One critical item must be addressed, and that is to clarify how the deletion mutants were verified. Whole genome sequencing is by far the preferred method; however, Sanger sequencing in combination with complementation (as was carried out) could be sufficient.

Response: We apologize for omitting this important piece of information. The deletion methods of *L. pneumophila* is as used in the ref 51. *L. pneumophila* strains containing in frame deletion were constructed through allelic exchange using a conditionally replicating plasmid pSR47s containing Kan^r and *sacB* gene (sucrose sensitivity marker). Sequences adjacent to the start and stop codons of each gene were amplified in two PCR reactions using $\Delta 3$ A1/ $\Delta 3$ A2 and $\Delta 3$ B1/ $\Delta 3$ B2 primer pairs (primer sequences are provided in Table S2), respectively. These were then cloned into the suicide plasmid pSR47s (Kan^r *sacB*) and introduced into *L. pneumophila* by triparental mating. After selecting for integration by plating onto kanamycin-containing medium, transconjugants were purified and plated onto 5% sucrose CYET plates to screen sucrose-resistant recombinants that had lost the plasmid. Standard colony PCR with primer pairs complementary to the flanking regions of deleted genes was used to verify deletions. The verification method is illustrated in the figure below.

Colony PCR using primers A1 and B2

If deletion was NOT successful, the PCR product would be 2.4 kb plus the size of deleted genes

If deletion was successful, the PCR product would be 2.4 kb

The information on constructing the $\Delta 3$ mutant strain and its verification has been added to lines 397–407 in the revised manuscript: “To construct Δ lpg2368- Δ lpg2369- Δ lpg2370 triple deletion mutant strain of *L. pneumophila* (termed $\Delta 3$), we first constructed deletion plasmid by cloning the upstream and downstream flanking regions into pSR47S. Briefly, the 1.2 kb fragment located upstream of *lpg2368* and the 1.2 kb fragment located downstream of *lpg2370* were obtained by PCR using $\Delta 3$ A1/ $\Delta 3$ A2 and $\Delta 3$ B1/ $\Delta 3$ B2 primer pairs (Table S2), respectively. The amplified PCR products were used as templates to produce a DNA fragment containing the flanking regions by fusion PCR with primers $\Delta 3$ A1/ $\Delta 3$ B2. After digestion with BamHI and Sall, the DNA fragment was inserted into BamHI/Sall-digested pSR47S⁵⁰. The deletion plasmid was then introduced into Lp02 by triparental mating, and conjugants were selected on CYET plates containing kanamycin (20 μ g/mL) and streptomycin (50 μ g/mL). Deletion mutants were verified by standard colony PCR using primers $\Delta 3$ A1/ $\Delta 3$ B2 from colonies grown on CYET plates containing 5% sucrose⁵¹”.

2. In addition, the size exclusion data for analyzing which protein(s) interact with others is unclear, as no sizes or calibrations are indicated. Presenting each sample alone, then adding in other samples, would clarify this. As it stands, no sound conclusions can be drawn from the data as shown.

Response: The samples used in the SEC analysis of interactions are now explained in detail and the results of pull-down assay for analyzing interactions between the proteins are provided in the Fig. s4. We hope these additional data can make our results more convincing. Please see line 175-183.

Strongly suggested changes

3. Include the organism name in the title to direct readers between different systems but with the same names.

Response: Following the reviewer's suggestion, the title was changed to "Molecular mechanism of toxin neutralization in the HipBST toxin-antitoxin system of *Legionella pneumophila*".

4. One of the most exciting findings is the use of this toxin as an effector; please move Fig S11 into the main text, and consider adding this information to the manuscript title.

Response: We thank the reviewer for interest in our findings on the role of HipT_{Lp} toxin (Lpg2370) as a T4SS effector protein. Fig. S11 was moved into the main text. We also attempted to add this information to the manuscript title, but we ultimately decided against it because it would make title rather long and/or confusing.

5. A pervasive statement throughout the manuscript (starting in the abstract), is that the HipT and HipS sequences "derive from" HipA. However, there is no data to support which direction evolution has gone, if at all – it is incorrect to imply one arose from the other or vice versa. It is worth remarking that they share consensus domain structure, sequence, and apparent function, but it should not be phrased with any specific relationship.

Response: Thank you for this helpful and insightful comment. We removed the statement that the HipT and HipS sequences derive from HipA throughout the manuscript, instead remarking their shared domain structure, sequence, and apparent function. Please see lines 23, 64–65, 167-168, and 295 of the revised manuscript.

6. The manuscript is more difficult to follow due to nomenclature of the proteins of interest. It would be helpful to establish early in the manuscript that Lg2370 is HipT, Lg2369 is HipS, and Lg2368 is HipB, then refer to them by these more descriptive names (with the Lg subscript). (I had to use a "cheat sheet" with 70=toxin, etc, to help me understand the manuscript).

Response: The authors apologize for the confusion caused by the nomenclature of the proteins in this study. We only started referring to Lpg2370, Lpg2369, and Lpg2368 as HipT_{Lp}, HipS_{Lp}, and HipB_{Lp}, respectively, at the end of the "Lpg2368-Lpg2369-Lpg2370 constitute the tripartite HipBST TA system" subsection of the Results section because it did not appear reasonable to utilize the HipBST TA system nomenclature until we provided experimental results that validate the proteins of interest as members of a HipBST TA system. Nonetheless, we do agree that the issues with nomenclature make the manuscript more difficult to follow. Therefore, we also established that Lpg2370 is HipT_{Lp}, Lpg2369 is HipS_{Lp}, and Lpg2368 is HipB_{Lp} in the last paragraph of the revised Introduction section (lines 82–83).

Other comments

Abstract

7. Line 17-18: based on the modifications of the text, I recommend changing "have been proposed" to "have been demonstrated", as the data behind those findings are rarely (ever?) questioned. Also, perhaps you can drop "important", leaving it as TA systems have "roles".

Response: The authors followed the reviewer's suggestions and rephrased the sentence

in lines 17–18.

8. Line 21-25: recommend splitting this into two (or more) sentences expressing one main idea each

Response: The original sentence was split into two sentences: “We found that together with two upstream genes *lpg2368* and *lpg2369*, *lpg2370* constitutes the recently discovered tripartite toxin-antitoxin system HipBST. Notably, the toxin *Lpg2370* (HipT_{Lp}) and the antitoxin *Lpg2369* (HipS_{Lp}) correspond to the C-terminal and N-terminal regions of the *E. coli* HipA from the type II TA system HipBA, respectively.”

Text

9. Line 35: it might be worth noting the TA systems are enriched in mobile genetic elements.

Response: The sentence has been rephrased and it now states the following: “Toxin-antitoxin (TA) systems are bacterial and archaeal genetic modules enriched in mobile genetic elements and chromosomes that comprise two or more closely linked genes encoding a toxin protein and its cognate antitoxin.” Please see lines 34–35 of the revised manuscript.

10. Line 45: remove “in normal growth conditions” (not sure what that would be!)

Response: We followed the suggestion and removed the phrase.

11. Line 49: the idea that Type II systems are most prevalent is likely heavily biased by what has been searched for, so perhaps this should be phrased as “most studied”, or “most focused on” or something similar

Response: Considering the reviewer’s suggestion and that the prevalence and degree of study of type II systems and HipBA modules are rather irrelevant to the topic of our study, we decided not to mention them. The sentence now states the following: “In type II TA systems such as HipBA modules, toxin neutralization depends on direct binding of a proteinaceous antitoxin.”

12. Line 50: needs a citation. Is HipA really the most prevalent?

Response: We thank the reviewer for pointing out this issue. For the reasons stated above, the authors decided to remove this sentence from the manuscript.

13. Line 61: please add “of the toxin” after the statement about allosteric regulation to clarify you are not referring to any action of transcription repression (conditional cooperativity)

Response: The phrase “of the toxin” was added after the statement about allosteric regulation. Please see line 56 of the revised manuscript.

14. Line 66: do both references 22 and 23 apply to this specific work discussed?

Response: HipBST module was first experimentally verified by the reference 23, whereas the HipA-like kinases were phylogenetically analyzed in reference 22. The reference 22 is therefore not cited in this sentence anymore.

15. Line 70: do not say “derived from” unless there is evidence this is the case

Response: The sentence was rephrased following the reviewer's comment and now states the following: "Importantly, the toxin HipT and the antitoxin HipS of the HipBST system were found to correspond to the N-terminal subdomain 1 and the core kinase domain of the *E. coli hipA* gene." Please see lines 64–65 of the revised manuscript.

16. Line 74: some of these sections are phrased strangely. The mechanism of toxin neutralization may have been unknown until now, but it is phrased as if it remains unknown today. The strangeness of this is because it is what is being reported, and therefore should not be considered unknown anymore.

Response: The sentence was revised to state the following: "However, the general mechanism for toxin neutralization in HipBST TA systems is not fully elucidated."

17. Please ensure the BioRxiv paper is discussed in this section and the presented results are framed in relationship to what is reported. I appreciate that it is now included in the Discussion, but should really also appear in the Introduction section

Response: Following the reviewer's suggestion, we added a few sentences to the Introduction section regarding the results reported in the bioRxiv paper and their relationship to our study. Please see lines 67–70 of the revised manuscript.

18. Lines 81-82: another strange phrase of "characterization is still pending" while this manuscript is reporting this characterization, so it is not "still pending".

Response: The sentence was revised to: "One such effector is the recently identified Lpg2370²⁵, which was previously predicted to be an E3 ligase but has not been characterized²⁶."

19. Line 82: remove the word "detectable" (for example, the opposite is not true – there is no such phrase as "undetectable sequence identity")

Response: The word detectable was removed and the sentence was rephrased (line 78 of the revised manuscript).

20. Line 83: and throughout please check the use of "terminal" and "terminus". I think in this line you are referring to it as a noun, so the "terminus"; if referring to a location it would be "terminal".

Response: The use of "terminal" and "terminus" was revised throughout the manuscript. Thank you for pointing out this issue.

21. Line 113: If HipA is the most prevalent family, why are the comparisons limited to the *E. coli* HipA?

Response: Although the HipBA TA systems are widespread in bacteria and archaea, HipBA was firstly identified in *E. coli*¹. The *E. coli* HipBA is also most studied and most well-understood on functional and molecular levels²⁻¹⁰. Besides, the HipBA TAs are quite conserved, so the *E. coli* HipBA was often used to be compared by the previous studies^{11,12} and we also compared Lpg2370 to HipA of *E. coli* HipBA in this study.

22. Line 131: These results demonstrate this is a Ser/Thr kinase, but are limited to auto-phosphorylation based on the presented data. I appreciate that the phosphorylated substrates may remain unknown, but I am not sure you can state it is a functioning Ser/Thr kinase without showing this functional data. Please clarify this in the text.

Response: The authors thank the reviewer for this helpful comment. Indeed, we only

demonstrated that Lpg2370 (HipT_{Lp}) shares notable sequence identity with the Ser/Thr kinase HipA from *E. coli* and that it can carry out autophosphorylation on the residue Ser54 within putative P-loop, but the potential substrates in *L. pneumophila* and/or host cell are still unknown. Thus, the sentence (line 126 of the revised manuscript) was rephrased as follows: “Taken together, these results suggest that Lpg2370 is a Ser/Thr kinase, though its substrates are currently unknown.”

23. Line 145: Why then would a prediction rely on such a small match, and why does this segment match FANCL?

Response: We checked the original alignment of Lpg2370 and FANCL (PDB: 4CCG) sequences reported in the Figure S6 of the Lin et al. study (the figure is provided below). Indeed, the alignment is based on a rather small sequence segment, yet this led several following studies to refer to Lpg2370 as a novel E3 ligase¹³⁻¹⁶. We hope our study can promote better understanding of Lpg2370 and its function.

```
ss_pred      CccccccCCCCCC-----cHHHHHHHcCC
Lpg2370      4 CPITYEKISDQEN-----YSQRLHLLSPQ 28
4ccg_X      307 CGICYAYQLDGTIPDQVCDNSQCGQPFHQICLYEWLRG 344
ss_dssp      CTTTCCSCBTTBCCCEECSCTTTCCEECHHHHHHHTT
```

Figure S6 (Sequence alignment of Lpg2370 and FANCL)¹⁷.

24. Line 148-149: are the resolutions of these compared structures comparable? It is easy to image a higher resolution structure such as in the current work would show more ordered regions.

Response: The authors confirmed that the resolutions of the compared structures are comparable: the resolutions of the crystal structure of the unphosphorylated HipA_{S0} (PDB ID:3TPB) is and HipA_{Ec} (PDB ID: 3DNU) are 1.88 Å and 1.54 Å, respectively. It should be noted that P-loop is disordered in the structures despite their high resolutions.

25. Line 168: this is a bit clunky “HipA C-terminal domain-containing proteins”. Can you come up with a better way to express this?

Response: The phrasing was changed to “proteins containing the C-terminal domain of HipA”. We hope that this expression will be less clunky.

26. Line 187: Based on these data you cannot exclude Lg2368 binding to either individual protein unless you specifically tested this. (See note below about the Figure for these data)

Response: We thank the reviewer for this comment. We performed pull-down assays to verify whether Lpg2368 binds to either Lpg2369 or Lpg2370 individually and further confirmed that Lpg2368 can only bind to the preformed Lpg2379-Lpg2369 complex. Associated experimental data were added to the Fig.s4 in the revised manuscript. Please see line 175-182.

27. Line 203: Why show the delta2 data with H/A mutant then?

Response: The authors thank the reviewer for bringing this point to our attention. We originally used the $\Delta 2$ deletion strain to investigate whether the endogenous expression of Lpg2370 is toxic to *L. pneumophila*. However, upon reinspection of our data, we concluded that $\Delta 2$ data is redundant, so it was removed from the manuscript.

28. Line 206: could you clarify? Is this the three individual open reading frames, and then the 68-69 ORFs together?

Response: *lpg2370* was cloned into the low-copy-number IPTG-inducible vector pZL507 to induce the expression of Lpg2370, whereas *lpg2368*, *lpg2369*, or *lpg2368-lpg2369* ORFs together were inserted into the plasmid pJL03 with the arabinose-inducible pBAD promoter. This way the roles of Lpg2369, Lpg2368 and the combination of Lpg2369 and Lpg2368 was assessed in neutralization of Lpg2370-mediated toxicity.

29. Line 208: just a typo, "0" instead of "delta" symbol

Response: The typo was corrected.

30. Line 212: What does "revert" convey?

Response: We apologize for ambiguous phrasing. The sentence was revised to explain our experimental results more clearly (lines 207–209 of the revised manuscript): "Co-expression of Lpg2368 and Lpg2369 was also found to counteract Lpg2370-dependent growth inhibition, whereas the expression of Lpg2368 without Lpg2369 could not prevent the growth inhibition (**Fig. 3e**)".

31. Line 213: Why is this, "in contrast"? I do not think I understood the meaning of "revert" in the previous sentence. Does this indicate that 68+69 are toxic when expressed? If so, how does this fit the model of toxin-antitoxin?

Response: We apologize for ambiguous phrasing. The sentence was revised to explain our experimental results more clearly (lines 238–240 of the revised manuscript): "Growth inhibition caused by the expression of Lpg2370 was counteracted by co-expression of Lpg2369, suggesting that Lpg2369 functions as the antitoxin (**Fig. 3e**). Co-expression of Lpg2368 and Lpg2369 was also found to counteract Lpg2370-dependent growth inhibition, whereas the expression of Lpg2368 without Lpg2369 could not prevent the growth inhibition (**Fig. 3e**)".

32. Line 220: please clarify these are conformations in crystals, leaving open a possibility (even if unlikely) that in solution it may be different

Response: The sentence was revised to clarify that our observations were based on the comparison of the crystal structures of pHipT_{Lp} and *E. coli* HipA S150A mutant: "Comparison of the crystal structure of pHipT_{Lp} and the structures deposited in the PDB revealed that the autophosphorylated P-loop in HipT_{Lp} adopts an orientation similar to that of the P-loop in the crystal structure of *E. coli* HipA S150A mutant."

33. It seems very strange / out of place / that mid-way through the experimental system is switched to *E. coli* 0127 for toxicity / mutagenesis. Can the authors provide some transitional statement as to why *Ec* was more appropriate?

Response: To clarify why we are switching to the *E. coli* O127 mid-way through experiments, we added the following transitional statement: "HipT_{O127} TA is used to perform the growth inhibition assays due to the easy manipulation of *E. coli* compared to *L. pneumophila*". Please see lines 239–240 of the revised manuscript.

34. Line 261: does the BioRxiv paper have a structure of the complex?

Response: The bioRxiv paper reported the structure of the HipT-HipS-HipB heterotrimer. Since our study reports the structures of pHipT, the HipT-HipS complex, and the HipT-

AMP-PNP complex, we believe that the structural data in the BioRxiv paper and our study complement each other and provide better understanding of HipBST TA systems on the molecular level.

35. Line 264: Why is it necessary to insert an RBS for the toxin gene? Is it to ensure stoichiometrically equal expression?

Response: RBS was inserted between the ORFs of Lpg2369 and Lpg2370 to express a 6×His-tagged Lpg2369 and untagged Lpg2370. Assuming that one of the two proteins may be expressed in excess to the other, this way we were able to obtain Lpg2370-Lpg2369 complex by co-elution during the Ni-NTA purification step. The excessive or unbound Lpg2370 would just flow through the Ni-NTA column, whereas the excessive or unbound Lpg2369 could then be removed from the protein sample in the later stages of purification, i.e., size-exclusion chromatography. In summary, insertion of RBS ensured purification of the Lpg2370-Lpg2369 complex formed at appropriate stoichiometry (in this case 1:1 stoichiometry) while removing excess or unbound proteins.

36. Line 305: see note above about assigning origins of HipA versus HipT-HipS

Response: The sentence was revised according to the reviewer's remarks and it now states the following: "One of the most striking features of the HipBST TA systems is that the role of antitoxin is taken by HipS which corresponds to the N-terminal portion of HipA toxin from the *E. coli* HipBA system."

37. Line 310-315: If the toxin alone is phosphorylated, and the toxin co-expressed with the antitoxin is not phosphorylated, why would this lead to questions about HipB? Please clarify which samples were used for SEC that allows this conclusion to be drawn.

Response: The lines in question restate the results described in lines 178–185 of the revised manuscript: "The co-expressed 6×His-tag Lpg2369 and untagged Lpg2370 were co-eluted using Ni affinity chromatography, and the size-exclusion chromatography analysis revealed that the peak is shifted forward by 0.4 ml compared to the peak of Lpg2370 alone, suggesting Lpg2369 can interact with Lpg2370 (**Fig. 3c**). Moreover, size-exclusion chromatography indicated that Lpg2368 co-elutes with the co-expressed 6×His-tagged Lpg2369-Lpg2370 complex and binds to the Lpg2369-Lpg2370 complex assembled *in vitro* (**Fig. 3c**), which was then further confirmed by the pull-down assays (**Fig. s4**). These results suggest that Lpg2370 directly interacts with Lpg2369, whereas Lpg2368 binds to a stable Lpg2369-Lpg2370 complex".

38. Line 319: this seems unlikely to be a coincidence. Comparing a T-S complex to a B-A complex seems incorrect, since T-S is homologous to HipA.

Response: The authors agree that comparing a T-S complex to a B-A complex appears incorrect considering that T-S complex is homologous to HipA. However, the recently released structure of T-S-B complex from HipBST_{O127}, which is more comparable to a B-A complex, supports our analysis. Please see revised analysis in lines 309–310 of the revised manuscript.

39. Line 422: just a small typo, should be "identified" (not "unidentified")

Response: The typo was corrected. We thank the reviewer for thorough review and analysis of our manuscript.

Fig 3

40. Panel b, what is the yellow line denoting on the toxin genes?

Response: The yellow line denotes the relative location of the P-loop within the toxin genes. A label "P-loop" was added above the yellow lines in the revised Fig. 3b.

41. Panel c, this is confusing. Which fractions are on the gel? Which eluted peaks correspond to what? Without information about molecular weight and column calibration (or comparison to the elution of individual proteins), we cannot follow any conclusions drawn from these data. I cannot understand the interaction of HipB from these data.

Response: We apologize for confusing data. The elution peaks of the samples are indicated and labelled clearly in the revised Figure 3c (shown below). In addition, we provide the results of pull-down assays used to study protein interactions and support the SEC data Fig. s4.

42. Panel d, e, why does this His to Ala mutation block toxicity? (e.g., what is the catalytic or functional role of this His?). These panels are way too small; it is difficult to distinguish the individual lines in panel e. Why is the delta3 (Lpg2370+vector) shown three times? Are these replicates? Why not show some average?

Response: Based on the sequence and structural comparison of Lpg2370 to *E. coli* HipA, the conserved residue H199 is the crucial residue for its kinase activity, we surmised that the H199A mutation would similarly inactivate the kinase activity of Lpg2370. Our data also indicates that the Lpg2370 H199A (H/A) mutant does not inhibit bacterial growth, which led us to conclude that the kinase activity of Lpg2370 is strictly required for its toxicity. For clarity, we indicated in the revised manuscript that Lpg2370 H199A mutant is catalytically inactive.

Three groups of plasmids were used for the toxicity assays in *L. pneumophila*, namely pZL507-2370+pJL03-2368, pZL507-Lpg2370+pJL03-Lpg2369, and pZL507-2370+pJL03-Lpg2368+Lpg2369. IPTG (200 μ M) was added to induce the expression Lpg2370, whereas arabinose (1%) was added to induce the expression of Lpg2368, Lpg2369, or Lpg2368+Lpg2369. "Lpg2370+vector" represents the addition of IPTG but not arabinose to that only the expression of Lpg2370 was induced. We have changed the

labelling in Figure 3e for clarity.

Fig 4

43. Panel a, how many replicates?

Response: The presented data for binding affinity between pHipTLp and AMP-PNP is from a single ITC experiment. A sentence was added to the legend of Fig. 4a to clarify this.

44. Panel b, should clarify what bond lengths or ranges were used to assign these dashed lines as bonds.

Response: Thank you for this suggestion. The distances between ATP and atoms involved in hydrogen bonds are 2.7–3.3 Å, which is within hydrogen bonding range. The distance range is now described in the legend of Figure 4b.

Fig 6

45. In general the wording in the legend seems off; I also suggest you drop “complex” from the description, as you’ve already established that HipT and HipS are bound together.

Response: The legend was revised following the reviewer’s suggestion.

46. Panel g, I believe the top graph is called a “thermogram” rather than a “binding curve”?

Response: The legend for Fig. 4a was changed to “ITC thermogram demonstrated that the HipTLp-HipSLp complex does not display detectable affinity for AMP-PNP”.

References

- 1 Moyed, H. S. & Broderick, S. H. Molecular cloning and expression of *hipA*, a gene of *Escherichia coli* K-12 that affects frequency of persistence after inhibition of murein synthesis. *J Bacteriol* **166**, 399–403, doi:10.1128/jb.166.2.399-403.1986 (1986).
- 2 Bokinsky, G. *et al.* HipA-triggered growth arrest and beta-lactam tolerance in *Escherichia coli* are mediated by RelA-dependent ppGpp synthesis. *J Bacteriol* **195**, 3173–3182, doi:10.1128/JB.02210-12 (2013).
- 3 Correia, F. F. *et al.* Kinase activity of overexpressed HipA is required for growth arrest and multidrug tolerance in *Escherichia coli*. *J Bacteriol* **188**, 8360–8367, doi:10.1128/JB.01237-06 (2006).
- 4 Germain, E., Castro-Roa, D., Zenkin, N. & Gerdes, K. Molecular mechanism of bacterial persistence by HipA. *Mol Cell* **52**, 248–254, doi:10.1016/j.molcel.2013.08.045 (2013).
- 5 Kaspary, I. *et al.* HipA-mediated antibiotic persistence via phosphorylation of the glutamyl-tRNA-synthetase. *Nat Commun* **4**, 3001, doi:10.1038/ncomms4001 (2013).
- 6 Korch, S. B. & Hill, T. M. Ectopic overexpression of wild-type and mutant

- hipA genes in *Escherichia coli*: effects on macromolecular synthesis and persister formation. *J Bacteriol* **188**, 3826–3836, doi:10.1128/JB.01740-05 (2006).
- 7 Li, C., Wang, Y., Wang, Y. & Chen, G. Interaction investigations of HipA binding to HipB dimer and HipB dimer + DNA complex: a molecular dynamics simulation study. *J Mol Recognit* **26**, 556–567, doi:10.1002/jmr.2300 (2013).
- 8 Moyed, H. S. & Bertrand, K. P. hipA, a newly recognized gene of *Escherichia coli* K-12 that affects frequency of persistence after inhibition of murein synthesis. *J Bacteriol* **155**, 768–775, doi:10.1128/JB.155.2.768-775.1983 (1983).
- 9 Schumacher, M. A. *et al.* Role of unusual P loop ejection and autophosphorylation in HipA-mediated persistence and multidrug tolerance. *Cell Rep* **2**, 518–525, doi:10.1016/j.celrep.2012.08.013 (2012).
- 10 Schumacher, M. A. *et al.* Molecular mechanisms of HipA-mediated multidrug tolerance and its neutralization by HipB. *Science* **323**, 396–401, doi:10.1126/science.1163806 (2009).
- 11 Vang Nielsen, S. *et al.* Serine–Threonine Kinases Encoded by Split hipA Homologs Inhibit Tryptophanyl-tRNA Synthetase. *mBio* **10**, doi:10.1128/mBio.01138-19 (2019).
- 12 Zhao, Y. *et al.* The HipAB Toxin–Antitoxin System Stabilizes a Composite Genomic Island in *Shewanella putrefaciens* CN-32. *Front Microbiol* **13**, 858857, doi:10.3389/fmicb.2022.858857 (2022).
- 13 Pisano, A. *et al.* Revisiting Bacterial Ubiquitin Ligase Effectors: Weapons for Host Exploitation. *Int J Mol Sci* **19**, doi:10.3390/ijms19113576 (2018).
- 14 Luo, J., Wang, L., Song, L. & Luo, Z. Q. Exploitation of the Host Ubiquitin System: Means by *Legionella pneumophila*. *Front Microbiol* **12**, 790442, doi:10.3389/fmicb.2021.790442 (2021).
- 15 Mondino, S., Schmidt, S. & Buchrieser, C. Molecular Mimicry: a Paradigm of Host–Microbe Coevolution Illustrated by *Legionella*. *mBio* **11**, doi:10.1128/mBio.01201-20 (2020).
- 16 Lockwood, D. C., Amin, H., Costa, T. R. D. & Schroeder, G. N. The *Legionella pneumophila* Dot/Icm type IV secretion system and its effectors. *Microbiology (Reading)* **168**, doi:10.1099/mic.0.001187 (2022).
- 17 Lin, Y. H. *et al.* RavN is a member of a previously unrecognized group of *Legionella pneumophila* E3 ubiquitin ligases. *PLoS Pathog* **14**, e1006897, doi:10.1371/journal.ppat.1006897 (2018).